# Comparisons of performances of structural variants detection algorithms in solitary or combination strategy

De-Min Duan[1,2], Chinyi Cheng[1], Yu-Shu Huang[3], An-ko Chung[4], Pin-Xuan Chen[1], Yu-An Chen[1]*, Jacob Shujui Hsu[1,5]*, Pei-Lung Chen[1,3,5,6]*

1 Graduate Institute of Medical Genomics and Proteomics, College of Medicine, National Taiwan University, Taipei, Taiwan, 2 Division of Cardiology, Department of Internal Medicine and The Cardiovascular Medical Center, Taipei Tzu Chi Hospital, Buddhist Tzu Chi Medical Foundation, Taipei, Taiwan, 3 Graduate Institute of Clinical Medicine, College of Medicine, National Taiwan University, Taipei, Taiwan, 4 Department of Internal Medicine, National Taiwan University College of Medicine, Taipei, Taiwan, 5 Genome and Systems Biology Degree Program, Academia Sinica and National Taiwan University, Taipei, Taiwan, 6 Division of Endocrinology and Metabolism, Department of Internal Medicine, National Taiwan University Hospital, Taipei, Taiwan

* paylong@ntu.edu.tw (PLC); jacobhsu@ntu.edu.tw (JSH); yuanchen0716@ntu.edu.tw (YAC)

**Data Availability Statement:** All relevant data are within the manuscript and its Supporting Information files.

## Abstract

Structural variants (SVs) have been associated with changes in gene expression, which may contribute to alterations in phenotypes and disease development. However, the precise identification and characterization of SVs remain challenging. While long-read sequencing offers superior accuracy for SV detection, short-read sequencing remains essential due to practical and cost considerations, as well as the need to analyze existing short-read data-sets. Numerous algorithms for short-read SV detection exist, but none are universally optimal, each having limitations for specific SV sizes and types. In this study, we evaluated the efficacy of six advanced SV detection algorithms, including the commercial software DRA-GEN, using the GIAB v0.6 Tier 1 benchmark and HGSVC2 cell lines. We employed both individual and combination strategies, with systematic assessments of recall, precision, and F1 scores. Our results demonstrate that the union combination approach enhanced detection capabilities, surpassing single algorithms in identifying deletions and insertions, and delivered comparable recall and F1 scores to the commercial software DRAGEN. Interestingly, expanding the number of algorithms from three to five in the combination did not enhance performance, highlighting the efficiency of a well-chosen ensemble over a larger algorithmic pool.

## Introduction

Genomic structural variants (SVs) are substantial alterations in the genome, varying in size from 50 to mega base pairs (bp) [1]. These variants include unbalanced gains and losses of DNA segments as well as balanced rearrangements. SVs are commonly classified into five types: deletions (DELs), insertions (INSs), duplications (DUPs), inversions (INVs), and

**Funding:** This work was supported by the Ministry of Science and Technology, Taiwan, under grant [111-2314-B-002 -194 -MY3 and 111-2314-B-002 -243 -MY3]. The funders had no role in study design, data collection and analysis, decision to publish, or preparation of the manuscript.

**Competing interests:** NO authors have competing interests.

complex translocations (CTXs). Owing to their large size, SVs account for a greater proportion of heritable sequence differences between individuals compared to single nucleotide variants (SNVs), with SVs constituting 0.5–1% of the genome and SNVs approximately 0.1% [2]. SVs contribute to genetic diversity and evolution at both individual and population levels. They exert a more profound impact on gene functions and phenotypic changes compared to SNVs and short insertion-deletions (indels) and are associated with various human diseases [3]. Polymorphic SVs are major contributors to common traits, including widespread diseases, while rare germline SVs are responsible for many rare genomic disorders [4].

With the advent of next-generation sequencing (NGS), the detection of SNVs and small indels (<50 bp) has become highly accurate, with an error rate of less than 0.1%. Despite substantial advancements in numerous SV detection methods, their effectiveness in short-read sequences remains limited [5]. Sequencing-based methods employ several approaches to derive information about SVs from short-read sequencing data: read-pair (RP), read-depth (RD), split-read (SR), and assembly (AS) approaches [6, 7]. RP uses the discordant alignment features of paired-end reads that encompass or overlap an SV. RD utilizes the depth features of paired-end reads that overlap an SV, indicating changes in read depth due to the SV. SR uses split (soft-clipped) alignment features of single-end or paired-end reads that span a breakpoint (BP) of an SV. AS detects SVs by aligning contigs, assembled from the entire or unmapped sequencing reads, to the reference sequence.

Despite great growth and improvement in numerous SV detection methods over the years, their performance in short-read sequences remains constrained. Sequencing-based approaches exhibit a high rate of SV miscalling, irrespective of the strategy employed. To address the limitations of short-read sequencing, recent efforts have utilized long reads generated through single-molecule sequencing technology for SV detection in human samples, employing assembly (AS) and/or split-read (SR) approaches [8–10].

Recently, long-read technologies such as Oxford Nanopore Technologies (ONT) and PacBio HiFi sequencing have made remarkable advancements, becoming more accessible and offering higher accuracy. These improvements help overcome many of the limitations associated with short-read sequencing [11]. However, despite these advancements and cost reductions making long-read sequencing more feasible, substantial challenges remain. These include not only higher costs, scalability issues, and stringent requirements for sample quantity and quality [12, 13], but also important considerations for the vast amounts of existing short-read data. For instance, many biobanks contain extensive archives of short-read data, which continue to have a high demand for reanalysis. The UK Biobank, for example, has sequenced the whole genomes of 150,119 participants using Illumina NovaSeq sequencing [14]. Similarly, while the All of Us (AoU) project has re-sequenced some samples with long-read techniques, it has already released over 100,000 datasets from Illumina whole genome sequencing (WGS) [15], highlighting the ongoing reliance on short-read approaches.

Today, there has been substantial progress in short-read-based general-purpose SV detection, with numerous SV detection algorithms being published. Many projects used popular SV detection algorithms showed a relatively high accuracy [16–20]. Although sequencing-based methods theoretically detect any type of SV, no single algorithm can accurately and sensitively detect all types and sizes [6, 21, 22]. Several studies have highlighted the individual performance of various callers. For example, Varuni Sarwal et al. evaluated the performance of SV detection tools on mouse and human WGS data, focusing on DELs detection using a comprehensive set confirmed by polymerase chain reaction (PCR) and the GIAB benchmark set [23].

Evaluations of combination strategies, however, remain relatively scarce. Some studies evaluated the performance of combining the consensus of individual tools. Daniel L. Cameron et al. assessed 10 SV callers individually and in possible m-of-n ensembles using well-

referenced cell lines such as NA12878, HG002, CHM1, and CHM13 [24]. They found that while simple ensemble calling sometimes improves the performance of the best individual callers, it is highly sensitive to the specific callers chosen for the ensemble. Similarly, Kosugi et al. analyzed the performance of 69 SV detection algorithms and the effectiveness of combining pairs of these algorithms [25]. They observed that when pairs of SV detection algorithms were analyzed, SVs commonly identified by multiple algorithms exhibited higher precision but lower recall than those identified by a single algorithm.

In the benchmarking study assessing 46 SV callers published between 2009 and 2017 [24], DELLY [16], LUMPY [17], Manta [18], and GRIDSS [19] were identified as popular tools based on their citation counts in the Web of Science database. While SvABA [20] was not included due to its 2018 publication date, it has shown robust performance in detecting both germline and somatic SVs in other assessments [26].

Building on these, our study reviews six state-of-the-art SV detection algorithms, including five previously mentioned tools and a commercial caller DRAGEN, to establish effective SV detection strategies. DELLY combines read RP and SR methods for SV detection, integrating paired-end short reads data from different insert sizes. LUMPY uses a probabilistic representation of SV breakpoints, leveraging RP and SR analyses and clustering overlapping breakpoints to identify SVs. Manta employs a two-phase workflow for high parallelization, building a graph of break-end associations and processing it for variant hypothesis generation, assembly, scoring, and VCF reporting. GRIDSS detects SVs of any size using AS, RP, and SR methods, generating break-end contigs and using a probabilistic model to infer SVs. SvABA operates within local 25k base pairs assembly windows with 2k base pairs overlaps, efficiently detecting SVs from short-read sequencing data with low resource requirements.

We analyze the SV count, type, and length detected by each algorithm and conduct a comprehensive evaluation of their performance. Additionally, we proposed combination strategies that involved multiple agreement strategies, which requires the consensus of more than two tools, as well as union strategies that integrates the outputs of all selected algorithms. Our analysis demonstrates the union approach achieved higher recall rates compared to single algorithms and attained similar recall and F1 scores to the commercial software DRAGEN.

## Materials and methods

### Truth set

The GIAB v0.6 Tier 1 benchmark set [1] was obtained from ftp://ftp-trace.ncbi.nlm.nih.gov/ReferenceSamples/giab/data/AshkenazimTrio/analysis/NIST_SVs_Integration_v0.6/. The HG00154, HG00733, and NA19240 benchmark set was obtained from HGSVC2 [27] (https://www.internationalgenome.org/data-portal/data-collection/hgsvc2). The total number of SVs in the truth set for each benchmark was detailed in S5 Table. Variants of the SV types 'DEL' or 'INS' were evaluated separately to assess the performance of deletions (DELs) and insertions (INSs).

### SV detection algorithms

We selected SV detection algorithms capable of handling short-read WGS through multiple methods. From the literature, we selected five publicly available SV detection algorithms: DELLY [16], LUMPY [17], Manta [18], GRIDSS [19], and SvABA [20]. Additionally, we included the commercial SV detection tool, Illumina's DRAGEN (Dynamic Read Analysis for GENomics) (https://support.illumina.com/sequencing/sequencing_software/dragen-bio-it-platform/downloads.html), for performance comparison, both as a single algorithm and in combination with different algorithms.

## Calling and refinement of structural variations

We obtained the whole-genome paired-end short reads of HG002 (NA24385) from the National Institute of Standards and Technology (https://www.nist.gov), and HG00514, HG00733, and NA19240 from HGSVC2 (https://www.internationalgenome.org/data-portal/data-collection/hgsvc2). These raw reads were aligned to the GRCh37 (for HG002) or GRCh38 (for HG00514, HG00733, and NA19240) versions of the human reference genome using BWA-MEM, as the SV benchmark set was based on the references. Subsequently, the resulting BAM files were processed through various SV detection algorithms, utilizing their default parameters as recommended by the authors for calling SVs.

For the refinement, filtering, and deduplication, the variants underwent a systematic four-step process: 1) SV Annotation: we employed the R package [28] to annotate SV types and lengths. 2) Size Exclusion: SVs smaller than 50 bp were excluded. 3) Record Deduplication: In cases where some SV callers (GRIDSS, SvABA, LUMPY, and Manta) generated output files containing separate records for the start and end of the same SV, we removed the end position record, as shown in S1, Fig 4) Final Filtering: Only variants marked 'PASS' in the 'FILTER' field were retained for further analysis.

## Combination strategies of multiple algorithms

Individual SVs were annotated with the name of their respective SV detection algorithms in the 'INFO' field (e.g., caller = DELLY). For grouping, one or more variants from different SV callers were considered a neighbor group if they existed within the range of +/- 500 bp of 'POS' (start position) and shared the same 'SVTYPE.' When referring to candidate SVs within a neighbor group, we retained the variant with the smallest start position as the representative.

Merging variants was conducted in the following ways: a combination of three algorithms, DELLY, Manta, and GRIDSS, formed the three SV (III) combination group, while a combination of five algorithms, Manta, DELLY, GRIDSS, LUMPY, and SvABA, formed the five SV (V) combination group. The "multiple agreement" strategy defined a 'true-called variant' if the members of a neighboring SV group were detected by at least two or more algorithms. Conversely, the union strategy defined a "true-called variant" if at least one member of a neighbor SV was detected by at least one algorithm, as demonstrated in S2 File.

## Evaluation of performance for single algorithm and combination strategies

The recall, precision, and F1 score for different single algorithms and combinations of multiple algorithms, either in "multiple agreement" strategy or union strategy, were calculated. We focused on the SV types 'DEL' and 'INS' to analyze the performance of different single algorithms and various combination strategies, because the truth sets only provide benchmarks for these types of SVs [1]. Variants detected by single algorithms and combination strategies (referred to as 'TEST' sets) were compared against the truth sets.

For evaluation, the true positive (TP) was defined if a variant in the truth set was located within the range of +/-500 bp of the 'POS' (start position) of a variant in the 'TEST' set and shared the same SV type. False positives (FP) were calculated as the total variant count of the 'TEST' set minus TP, and false negatives (FN) were calculated as the total variant count of the 'TRUTH' set minus TP. Precision, recall, and F1 score were then calculated using the following formulas.

$$\text{precision} = \frac{\text{TP}}{\text{TP} + \text{FP}} \tag{1}$$

$$recall = \frac{TP}{TP + FN} \tag{2}$$

$$F1 = 2 \times \frac{precison \times recall}{precision + recall} \tag{3}$$

To determine the rank of overall performance of each single caller and combination strategies, combined precision score (cPr), combined recall score (cRc), and combined F1 score (cF1), in which the values for all four data sets were integrated, and were calculated by micro average as follows:

$$cPr = \frac{TP_1 + TP_2 + TP_3 + TP_4}{TP_1 + TP_2 + TP_3 + TP_4 + FP_1 + FP_2 + FP_3 + FP_4} \tag{4}$$

$$cRc = \frac{TP_1 + TP_2 + TP_3 + TP_4}{TP_1 + TP_2 + TP_3 + TP_4 + FN_1 + FN_2 + FN_3 + FN_4} \tag{5}$$

$$cF1 = 2 \times \frac{cPr \times cRc}{cPr + cRc} \tag{6}$$

## Computational resources

All bioinformatic computations were conducted on the Taiwania 3 computational platform of the Taiwan National Center for High-Performance Computing (NCHC), recognized as the most powerful CPU high-performance computing server available for open- service applications in Taiwan. A Python script was developed to refine, filter, merge, intersect, and union the variants, enabling the calculation of recall, precision, and F1 score for different single algorithms and various combination strategies.

## Ethics approval and consent to participate

This research project solely utilizes public datasets. The whole-genome paired-end short reads of HG002 were acquired from the National Institute of Standards and Technology (NIST) website (https://www.nist.gov). Additionally, the GIAB v0.6 Tier 1 benchmark set was obtained from ftp://ftp-trace.ncbi.nlm.nih.gov/ReferenceSamples/giab/data/AshkenazimTrio/analysis/NIST_SVs_Integration_v0.6/. The whole-genome paired-end short reads and benchmark sets of HG00154, HG00733, and NA19240 were obtained from HGSVC2 (https://www.internationalgenome.org/data-portal/data-collection/hgsvc2). As such, the study does not involve direct participation of human subjects. Therefore, formal ethics approval and consent to participate are not applicable.

## Results

### SV detection methods included in this study

We first reviewed previous literature on short-reads SV callers, and evaluated criteria such as superior performance, widespread recognition, active development, and software/documentation quality, as well as their ease of use on our system. Ultimately, we selected five SV detection algorithms that demonstrated good performance in recent benchmarking studies [25, 26, 29, 30], namely DELLY [16], LUMPY [17], Manta [18], GRIDSS [19], and SvABA [20]. To provide a comprehensive representation of different SV detection methods, these algorithms were

**Table 1. Overview of SV detection methods and the types of SVs identified by different tools.**

| SV caller | Version | Algorithm | Published year | First author | Latest updated year | Tools webpage |
|---|---|---|---|---|---|---|
| Manta [18] | v1.6.0 | RP-SR-AS | 2016 | Xiaoyu Chen | 2019 | https://github.com/Illumina/manta |
| DELLY [16] | v1.1.5 | RP-SR | 2012 | Tobias Rausch | 2024 | https://github.com/DELLYtools/DELLY |
| GRIDSS [19] | v2.13.2 | RP-SR-AS | 2017 | Daniel L Cameron | 2023 | https://github.com/PapenfussLab/GRIDSS |
| LUMPY [17] | v0.3.1 | RP-SR-RD | 2014 | Ryan M Layer | 2024 | https://github.com/brentp/smoove |
| SvABA [20] | v1.1.0 | RP-SR-AS | 2018 | Jeremiah A Wala | 2024 | https://github.com/walaj/svaba |

Input: BAM; Output: VCF (Variant calling format). RP: read-pair; SR: split read; RD: read depth; AS: assembly.

chosen based on their proven performance. Additionally, we included the commercial SV detection software DRAGEN to further enrich our analysis. Callers were run using the recommended parameters and results were obtained.

These selected algorithms share a common feature—they can handle short-read whole WGS based on multiple methods and require a BAM file of the mapped reads. Most callers utilize more than one programming language and leverage multi-core architectures. Table 1 outlines their dependencies and their ability to detect specific types of SVs.

## Evaluation of SV detection methods based on the GIAB benchmark set and well-referenced cell lines

We next set out to determine the overall performance of SV callers using the sample HG002, for which a high-confidence SV call set aligned to the GRCh37 version has been established as a standard for SV evaluation [1]. This evaluation is based on the genomic data of the son (HG002) in a broadly consented Ashkenazim trio, provided by the Genome in a Bottle Consortium (GIAB). We obtained the whole-genome paired-end short reads of HG002 from the National Institute of Standards and Technology (https://www.nist.gov). Additionally, to enhance the genetic diversity of the samples and provide a more comprehensive assessment, we obtained benchmark sets from HGSVC2 [27] which contains a catalogue of SVs, including INSs and DELs, identified across more than 3,000 genomes from the 1000 Genomes Project. Three other well-referenced cell lines HG00514, HG00733, and NA19240 were also obtained from HGSVC2 in our analysis.

Based on the reference settings consistent with the truth sets, the raw reads underwent alignment to the GRCh37 (for HG002) and hg38 (for HG00514, HG00733, and NA19240) version of the human reference genome using BWA-MEM followed by our analysis workflow (Fig 1). The resulting BAM files served as input for various SV detection algorithms, executed with their default parameters based on author recommendations. The six SV callers, utilizing different algorithms (Table 1), successfully identified all five types of SVs in the true set: DELs, INSs, DUPs, INVs, and CTXs (Fig 2 and S1–S4 Tables).

The detection performance of the six SV callers varied, as shown in Fig 2 and S1–S4 Tables. For HG002, DRAGEN exhibited the highest number of detected SVs (10,912), followed by Manta (7,758), DELLY, LUMPY, SvABA, and GRIDSS. The other three datasets obtained from HGSVC2 showed a similar pattern in the number of detected SVs for each caller, as seen in HG002 (Fig 2A), except for LUMPY, which detected a higher number of CTX SVs in the HGSVC2 datasets.

We also analyzed the distribution of SV sizes detected by various algorithms. Notably, the SV size distribution varied among the different algorithms (Fig 2B and S1–S4 Tables). GRIDSS identified a large number of small indels (S1–S4 Tables, S3 Fig), while DELLY and LUMPY, despite both employing SR and RP algorithms, showed variations in their detected SV sizes

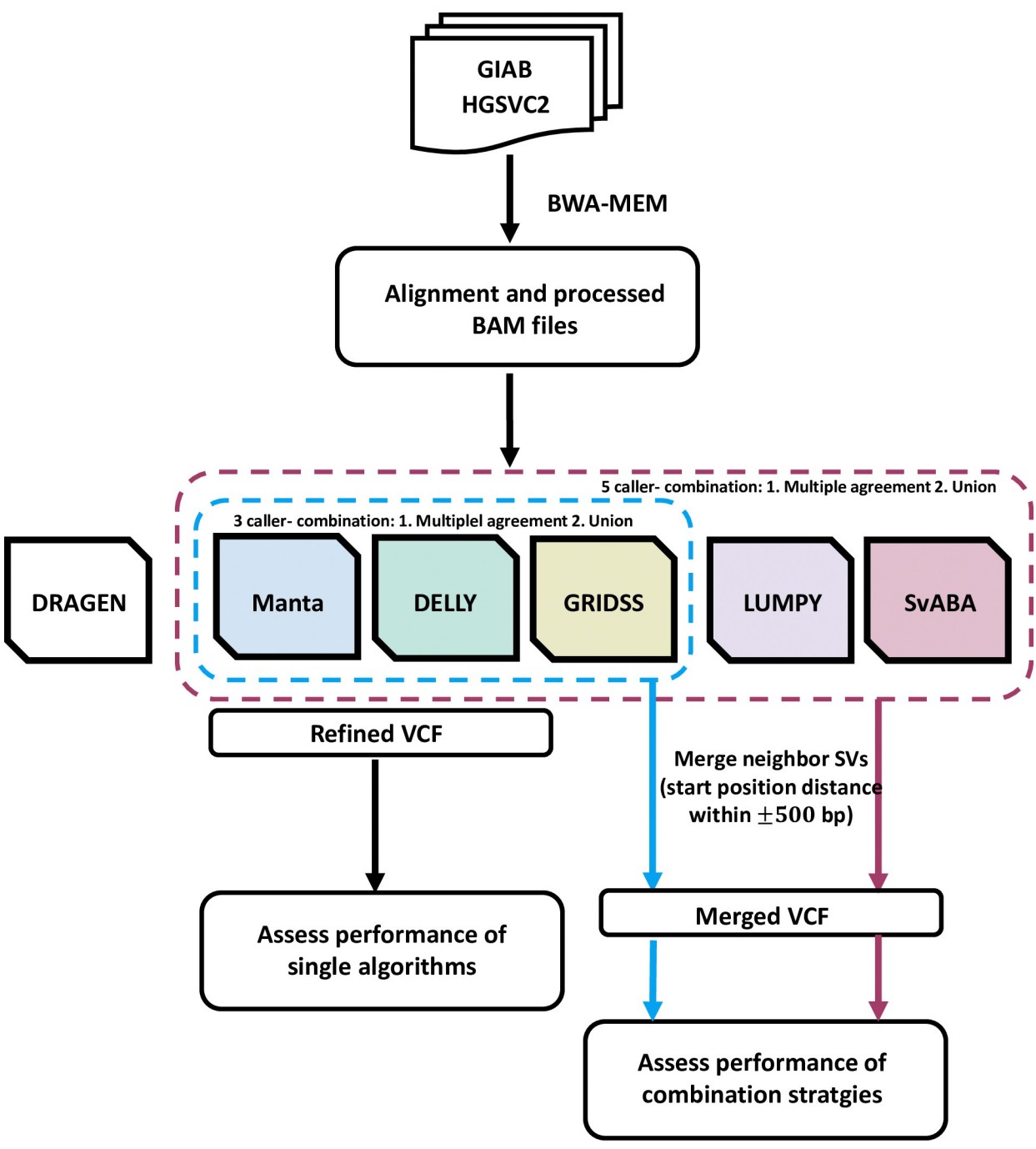

**Fig 1. SV detection workflow.** SV detection workflow starts with benchmark set collection from GIAB and HGSVC2, followed by sequence alignment using BWA-MEM to produce BAM files. These BAM files are then processed by various SV callers (DRAGEN, Manta, DELLY, LUMPY, GRIDSS, SvABA) to generate VCF files, which are subsequently used for performance assessment, including single tool and combination strategies, based on recall, precision, and F1 score.

and types. The SVs identified by DRAGEN, Manta, and SvABA primarily ranged from 50 to 10K base pairs (Fig 2B and S1–S4 Tables). Each algorithm detected DELs, DUPs, and INVs. Although all algorithms identified INSs, LUMPY and SvABA showed particularly poor performance in detecting these variants. CTXs were detected by all algorithms except DELLY

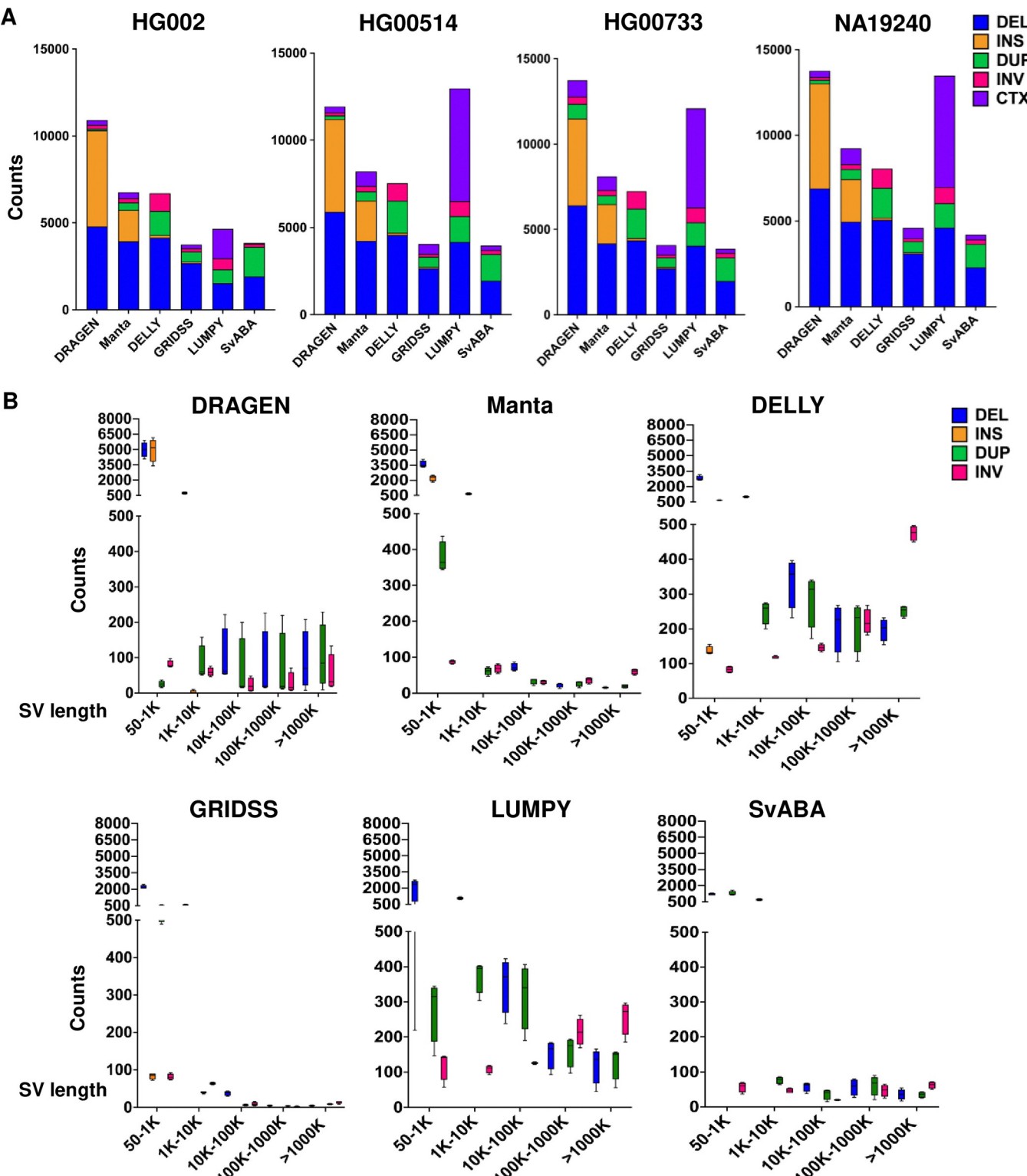

**Fig 2. The SVs detected by six SV callers. (A)** SVs of sizes ≥ 50 bp and CTX detected by individual SV callers in samples, HG002, HG00514, HG00733, and NA19240. **(B)** Length distribution of different variants for all samples detected by individual SV callers. The maximum, minimum, and median are based on the integrated values from all sample sets. DEL: deletion, INS: insertion, DUP: duplication, INV: inversion, and CTX: complex translocation.

(Fig 2 and S1–S4 Tables). The distribution of SV types was consistent across the datasets analyzed.

## The individual performance of each algorithm

To evaluate the recall, precision, and F1 score of each algorithm, we acquired the GIAB v0.6 Tier 1 benchmark dataset [1]. The Tier 1 benchmark regions, representing areas of the highest confidence, cover approximately 2.51 billion base pairs (Gbp) of genomic space. All variants within these regions have been verified and confirmed by at least one diploid assembly. The benchmark datasets of HG00514, HG00733, and NA19240 were also obtained from HGSVC2 [27]. The distribution of SVs in this benchmark set predominantly falls in the range of 50 to 10k base pairs (Fig 3A and S5 Table). We assessed the distribution of SVs within the subsets of INSs and DELs. Since LUMPY and SvABA did not detect INSs in size larger than 50 base pairs (as shown in Fig 2 and S1–S4 Tables), our performance comparison treated DELs and INSs separately. Specifically, we evaluated the performance of all callers for DELs, whereas for INSs, LUMPY and SvABA were excluded.

We considered SVs if they were located within a range of +/-500 base pairs of the 'POS' (start position) and were of the same SV type as the truth set SVs. Fig 3 and Table 2 present the detailed performance of each single algorithm. In HG002, for DELs, DRAGEN performed the best, identifying 3,425 true positives (TPs), followed by Manta, DELLY, GRIDSS, and SvABA, with LUMPY identifying the fewest. As for total error (FP + FN), DRAGEN and Manta also had the lowest error rates, while LUMPY had the highest. Similar trends were observed across other HGSVC2 datasets (Fig 3B and Table 2).

For INSs, DRAGEN and Manta again led in TP counts, while DELLY and GRIDSS had lower FPs but unacceptably high false negatives (FNs) across all datasets (Fig 3B and Table 2).

For each distinct algorithm, we computed the recall, precision, and F1 score. As shown in Fig 3C and Table 2, for DELs across all benchmark sets, DRAGEN achieved the highest F1 scores (0.64–0.68), followed closely by Manta, DELLY, and GRIDSS. LUMPY scored the lowest F1 in HG002 (0.19), while SvABA had mid-range scores in HG002 but lower in other datasets. Among all non-commercial software, Manta had the highest recall (0.43–0.55%), with GRIDSS showing lower recall but higher precision (0.86–0.95).

We also conducted a performance analysis of INSs, but most single algorithms performed disappointingly. Across all datasets, DRAGEN achieved the highest F1 scores (0.45–0.55), followed by Manta. DELLY (0.02–0.04) and GRIDSS (0.01–0.02) had very unsatisfactory F1 scores. Among non-commercial software, Manta had the highest recall (0.16–0.19) but only moderate precision (0.77–0.96). In contrast, GRIDSS and DELLY showed better precision (0.91–0.99) but poor recall (0.00–0.02%). Overall, low recall impacted the performance of these tools (Fig 3C and Table 2).

The relative runtime and maximum memory usage of each individual SV caller were tested using three HGSVC2 datasets and are presented in S2 Fig. Total CPU time reflects the overall CPU usage, while wall time represents the actual elapsed time. Wall time can be shorter than total CPU time when parallelization is efficiently utilized. Among the non-commercial tools, Manta exhibited the lowest run time and maximum memory usage across all datasets, whereas GRIDSS required the most elapsed time and memory.

## Systematic identification of the performances of combination strategies

Due to the overall unsatisfactory results with single algorithms, we turned to a strategy commonly employed in previous studies [25, 26, 31]—selecting SVs commonly called by multiple algorithms to enhance the precision of the identified SVs. In our pursuit of improved SV

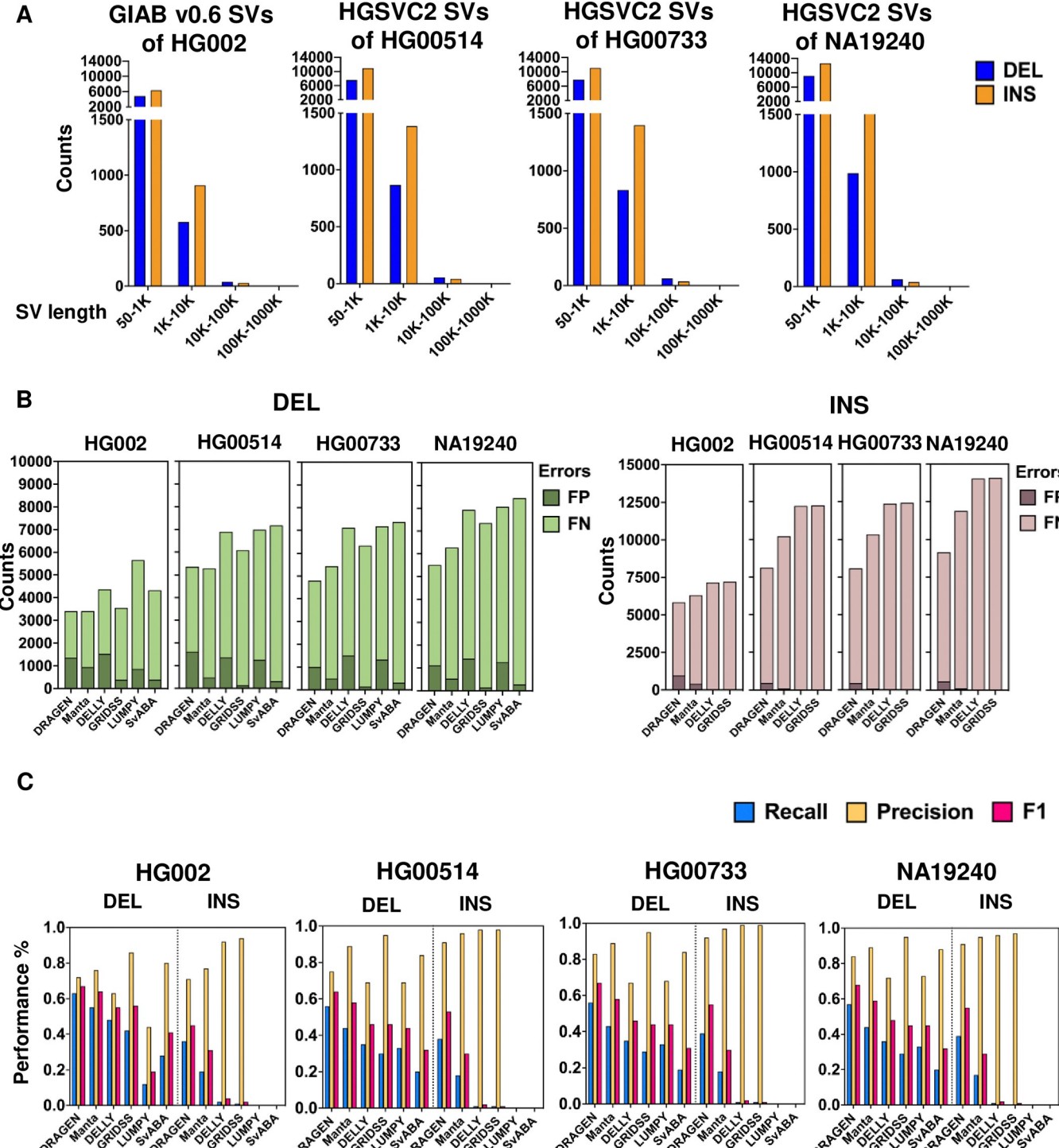

**Fig 3. The distribution of SVs in truth sets and the performance of individual algorithms.** (A) The distribution of SVs size and types in truth sets. (B) The comparison of False negative (FN) and False positive (FP) numbers among individual algorithms. (C) The precision, recall, and F1 score of each individual algorithm in detecting "DELs" and "INSs" with the size ≥ 50 bp. DEL: deletion, INS: insertion.

**Table 2. Performance of single algorithm in detecting DELs and INSs.**

| HG002 | | | | | | | |
|---|---|---|---|---|---|---|---|
| **SV caller** | **DEL** | **TP** | **FP** | **FN** | **Recall** | **Precision** | **F1** |
| DRAGEN | 4,787 | **3,425** | 1,362 | **2,039** | **0.63** | 0.72 | **0.67** |
| Manta | 3,935 | 2,996 | 939 | 2,468 | 0.55 | 0.76 | 0.64 |
| DELLY | 4,142 | 2,620 | 1,522 | 2,844 | 0.48 | 0.63 | 0.55 |
| GRIDSS | 2,681 | 2,299 | **382** | 3,165 | 0.42 | **0.86** | 0.56 |
| LUMPY | 1,528 | 668 | 860 | 4,796 | 0.12 | 0.44 | 0.19 |
| SvABA | 1,910 | 1,526 | 384 | 3,938 | 0.28 | 0.80 | 0.41 |
| **SV Caller** | **INS** | **TP** | **FP** | **FN** | **Recall** | **Precision** | **F1** |
| DRAGEN | 3,403 | **2,426** | 977 | **4,855** | **0.36** | 0.71 | **0.45** |
| Manta | 1,800 | 1,387 | 413 | 5,894 | 0.19 | 0.77 | 0.31 |
| DELLY | 155 | 142 | 13 | 7,139 | 0.02 | 0.92 | 0.04 |
| GRIDSS | 90 | 85 | **5** | 7,196 | 0.01 | **0.94** | 0.02 |
| **HG00514** | | | | | | | |
| **SV caller** | **DEL** | **TP** | **FP** | **FN** | **Recall** | **Precision** | **F1** |
| DRAGEN | 6,403 | **4,790** | 1,613 | **3,739** | **0.56** | 0.75 | **0.64** |
| Manta | 4,183 | 3,710 | 473 | 4,819 | 0.44 | 0.89 | 0.58 |
| DELLY | 4,359 | 2,996 | 1,363 | 5,533 | 0.35 | 0.69 | 0.46 |
| GRIDSS | 2,718 | 2,582 | **136** | 5,947 | 0.30 | **0.95** | 0.46 |
| LUMPY | 4,053 | 2,798 | 1,255 | 5,731 | 0.33 | 0.69 | 0.44 |
| SvABA | 1,981 | 1,668 | 313 | 6,861 | 0.20 | 0.84 | 0.32 |
| **SV Caller** | **INS** | **TP** | **FP** | **FN** | **Recall** | **Precision** | **F1** |
| DRAGEN | 5,100 | **4,654** | 446 | **7,679** | **0.38** | 0.91 | **0.53** |
| Manta | 2,291 | 2,204 | 87 | 10,129 | 0.18 | 0.96 | 0.30 |
| DELLY | 128 | 126 | 2 | 12,207 | 0.01 | **0.98** | 0.02 |
| GRIDSS | 90 | 88 | **2** | 12,245 | 0.01 | **0.98** | 0.01 |
| **HG00733** | | | | | | | |
| **SV caller** | | **TP** | **FP** | **FN** | **Recall** | **Precision** | **F1** |
| DRAGEN | 5,895 | **4,896** | 999 | **3,798** | **0.56** | 0.83 | **0.67** |
| Manta | 4,232 | 3,749 | 483 | 4,945 | 0.43 | 0.89 | 0.58 |
| DELLY | 4,574 | 3,075 | 1,499 | 5,619 | 0.35 | 0.67 | 0.46 |
| GRIDSS | 2,655 | 2,512 | **143** | 6,182 | 0.29 | **0.95** | 0.44 |
| LUMPY | 4,170 | 2,843 | 1,327 | 5,851 | 0.33 | 0.68 | 0.44 |
| SvABA | 1,944 | 1,634 | 310 | 7,060 | 0.19 | 0.84 | 0.31 |
| **SV Caller** | **INS** | **TP** | **FP** | **FN** | **Recall** | **Precision** | **F1** |
| DRAGEN | 5,313 | **4,868** | 445 | **7,627** | **0.39** | 0.92 | **0.55** |
| Manta | 2,312 | 2,236 | 76 | 10,259 | 0.18 | 0.97 | 0.30 |
| DELLY | 130 | 129 | 1 | 12,366 | 0.01 | **0.99** | 0.02 |
| GRIDSS | 81 | 80 | **1** | 12,415 | 0.01 | **0.99** | 0.01 |
| **NA19240** | | | | | | | |
| **SV caller** | **DEL** | **TP** | **FP** | **FN** | **Recall** | **Precision** | **F1** |
| DRAGEN | 6,897 | **5,775** | 1,122 | **4,390** | **0.57** | 0.84 | **0.68** |
| Manta | 4,955 | 4,423 | 532 | 5,742 | 0.44 | 0.89 | 0.59 |
| DELLY | 5,061 | 3,649 | 1,412 | 6,516 | 0.36 | 0.72 | 0.48 |
| GRIDSS | 3,098 | 2,957 | **141** | 7,208 | 0.29 | **0.95** | 0.45 |
| LUMPY | 4,608 | 3,354 | 1,254 | 6,811 | 0.33 | 0.73 | 0.45 |
| SvABA | 2,283 | 2,001 | 282 | 8,164 | 0.20 | 0.88 | 0.32 |

*(Continued)*

**Table 2.** (Continued)

| SV Caller | INS | TP | FP | FN | Recall | Precision | F1 |
|---|---|---|---|---|---|---|---|
| DRAGEN | 6,144 | **5,574** | <u>570</u> | **8,579** | **0.39** | <u>0.91</u> | **0.55** |
| Manta | 2,488 | 2,375 | 113 | 11,778 | 0.17 | 0.95 | 0.29 |
| DELLY | 134 | 129 | 5 | 14,024 | 0.01 | 0.96 | 0.02 |
| GRIDSS | 72 | <u>70</u> | **2** | <u>14,083</u> | <u>0.00</u> | **0.97** | <u>0.01</u> |

FN: false negative; FP: false positive; TP: true positive; DEL: deletion; INS: insertion. The best results are highlighted in bold and the worst results are highlighted with underlines.

calling accuracy, we conducted a thorough assessment of accuracy, precision, and F1 scores across various combination strategies, specifically exploring "multiple agreement" and union strategies involving three or five algorithms.

For our three-algorithm combination strategies, we selected Manta, DELLY, and GRIDSS, showcasing superior performance among all tools. Meanwhile, in the case of five-algorithm combination strategies, we considered all tools except the commercial DRAGEN. Each individual SV was annotated by the name of its SV detection caller in the 'INFO' field. Based on the original study of the GIAB benchmark set [1], the process to form the SV benchmark includes the criterion: "Filter Complex: If two or more supported variants ≥ 50 bp were within 1000 bp of each other, they were excluded because they are potentially complex or inaccurate." Therefore, we referenced this threshold of 1000 bp and set up a +/- 500 bp threshold for our analysis. Variants from different SV callers were grouped as neighboring SVs if they overlapped within the range of +/- 500 bp of 'POS' (start position) and shared the same 'SVTYPE.' The "multiple agreement" strategy categorizes variants as 'true-called' only if all members of the neighboring SV are detected by at least two algorithms. In contrast, the union strategy deems a variant as 'true-called' if at least one member of the neighboring SV is detected by at least one algorithm.

We evaluated the distance distribution of variant positions in neighboring SVs across different combination strategies (S1 Fig). The counts of neighboring SVs and the number of contributing member callers across different combination strategies are presented in S6 Table. All variants within a neighboring SV group were within 0 to 500 bp, as specified by our settings for combining structural variants. The median distance of neighboring SVs across sample sets was 4.0–5.0 bp for III-multiple agreement, 7.0–10.0 bp for V-multiple agreement, and 0 bp for both III-union and V-union. The interquartile range (IQR) for neighboring SV distances was 7.0 bp for III-multiple agreement, 17.0–23.0 bp for V-multiple agreement, 1.0–2.0 bp for III-union, and 7.0–10.0 bp for V-union, indicating that the majority of neighboring SV distances were very close.

The majority count of neighboring SVs within the three SV-caller group was fewer than three member variants, and the majority count of neighboring SVs within the five SV-caller group was fewer than five member variants (S6 Table). These indicate that very few distinct SVs were grouped due to the +/- 500 bp merge threshold.

The Venn diagram effectively illustrates the agreement among each algorithm in both three and five combination strategies (Fig 4). The distribution of SVs in different combination strategies is depicted in Fig 5A and S7 Table, presenting the total SV count across sample sets in the "multiple agreement agreement" and union strategies involving three or five algorithms.

For DELs in HG002, DRAGEN had the highest TP count (3,425), followed by V-union, III-union, V-multiple agreement, and the lowest for III-multiple agreement (2,554). The FP count was lowest for III-multiple agreement (533) and highest for V-union (2,199). Similar results

## A  Neighboring  SVs shared by III member calllers

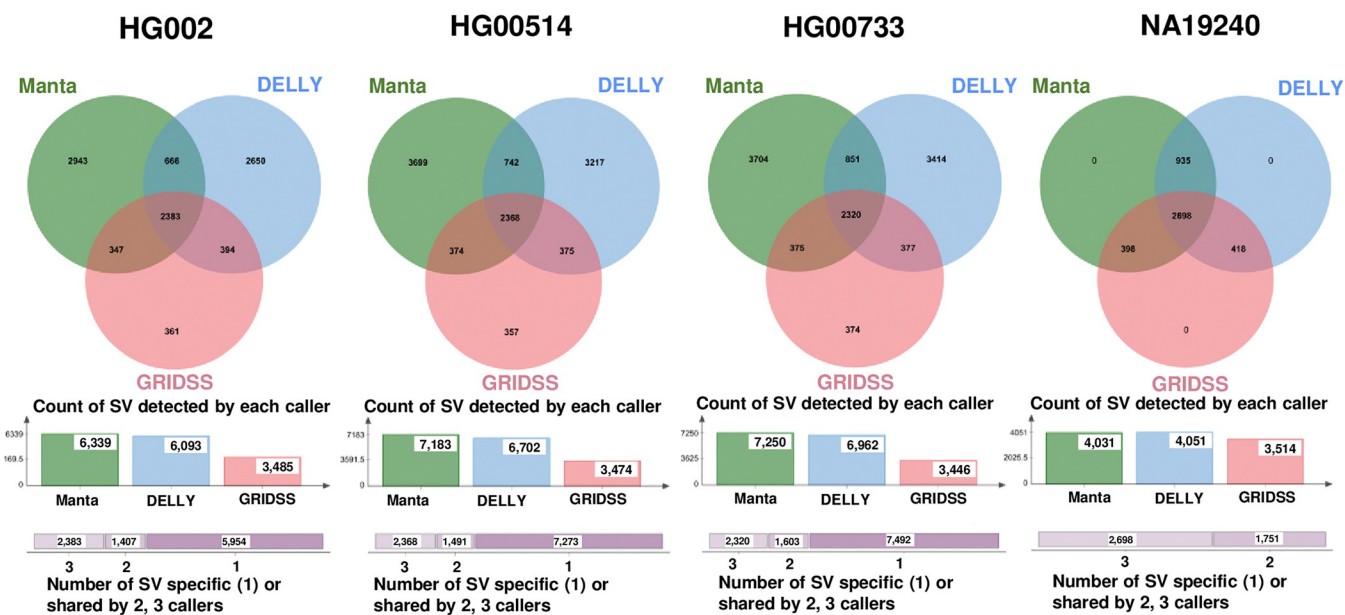

## B  Neighboring SVs shared by V member calllers

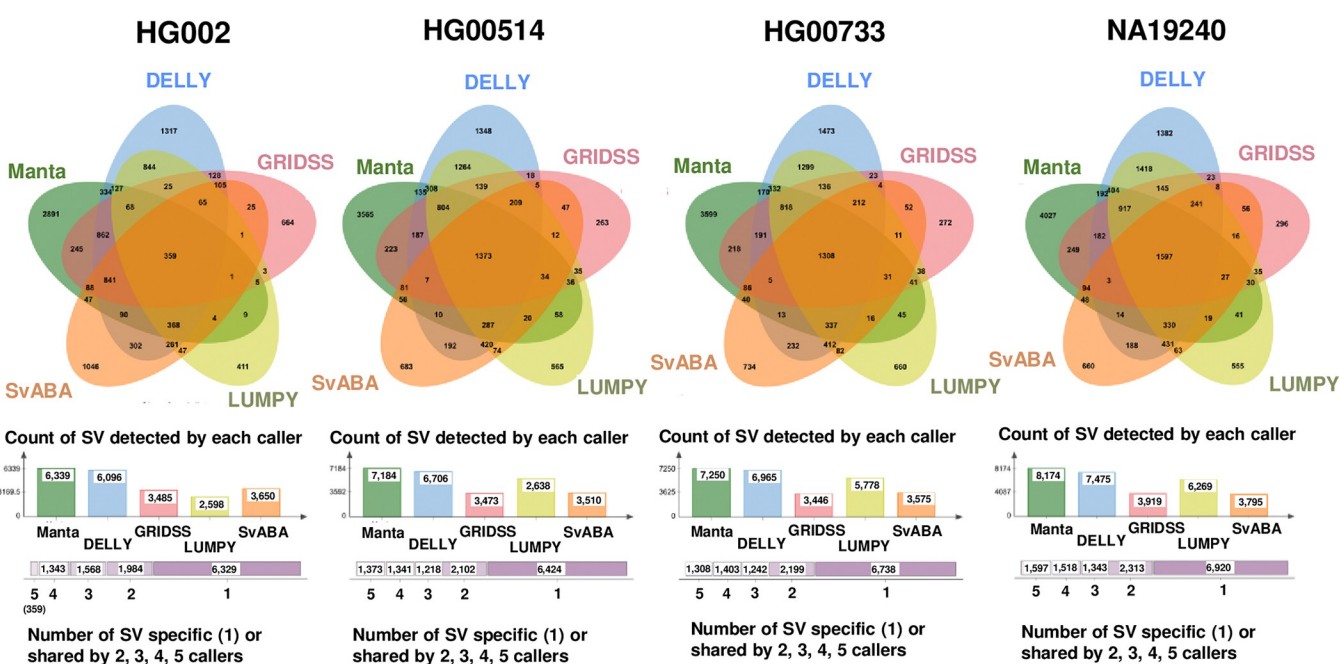

**Fig 4. Concordance of neighbor SVs detected by member callers in different combination strategies. (A)** The agreement among three SV detection tools (Manta, DELLY, and GRIDSS). **(B)** The agreement among five SV detection tools (Manta, DELLY, GRIDSS, LUMPY, and SvABA).

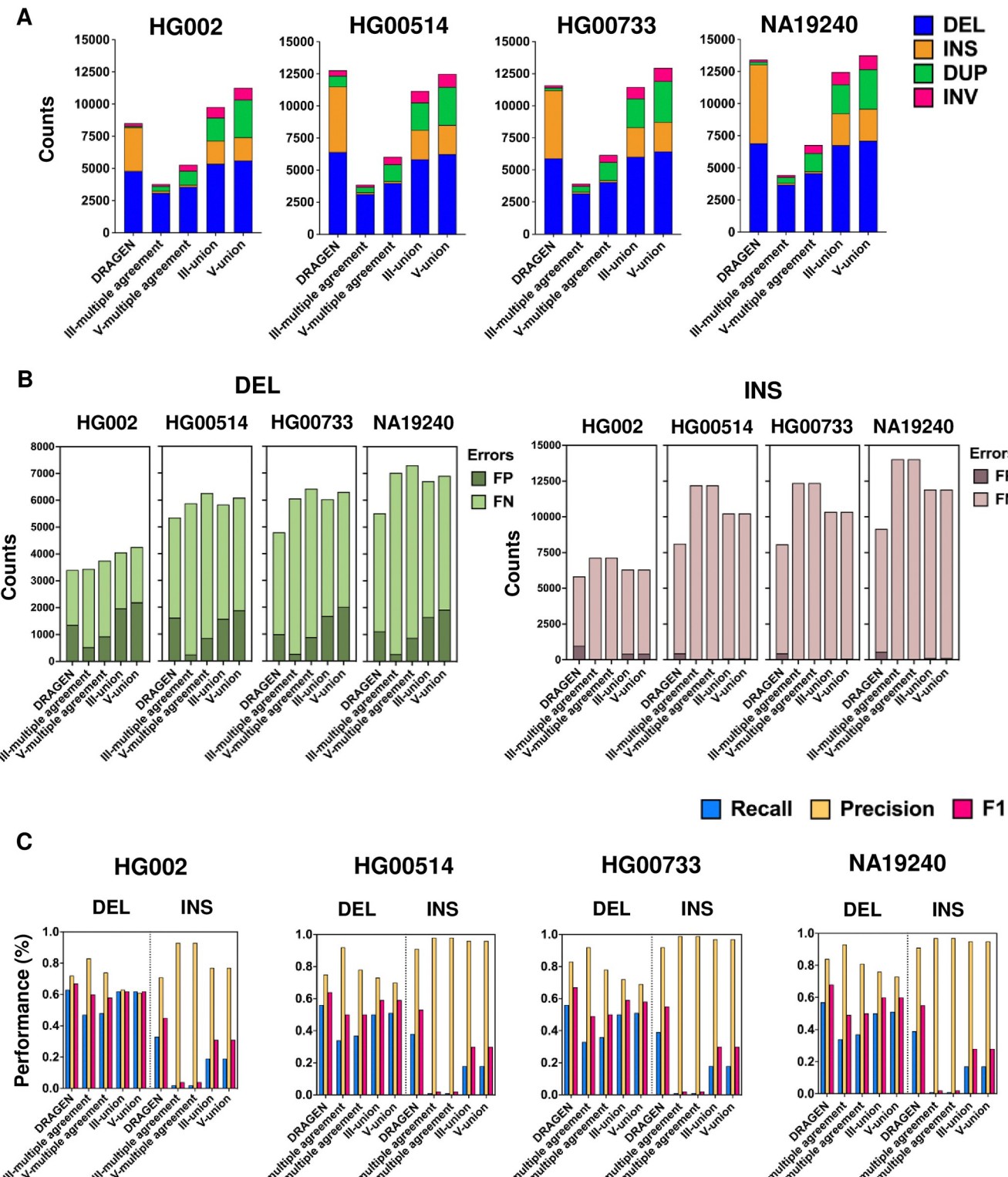

**Fig 5. Distribution and performance of combination strategies.** **(A)** The distribution of SVs in different combination strategies. **(B)** The comparison of False negative (FN) and False positive (FP) numbers among individual algorithms. **(C)** The precision, recall, and F1 score of combination strategies in the detection of DELs and INSs. DEL: deletion, INS: insertions, DUP: duplication, INV: inversion.

were observed in the HGSVC2 datasets. Regarding total errors (FP + FN), five-algorithm strategies generally produced more errors than three-algorithm strategies. For INSs, DRAGEN had the highest TP counts, while multiple agreement had the highest total errors. Overall, combination strategies still struggled with a high FN rate (Fig 5B and Table 3).

For each combination strategy, we calculated recall, precision, and the F1 score. As shown in Fig 5C and Table 3, for DELs across sample sets, DRAGEN achieved the highest F1 score (0.64–0.68), followed by III-union and V-union, with III-multiple and V-multiple agreement scoring lower. DRAGEN had the highest recall, closely followed by the union groups (0.50–0.62), while multiple agreement groups had lower recall (0.33–0.48). In contrast, precision was higher in the multiple agreement groups (0.74–0.93) and lower in the union groups (0.61–0.76).

For INSs, the performance of combination strategies was also disappointing across all benchmark sets (Fig 5 and Table 3). DRAGEN achieved the highest F1 score (0.45–0.55), followed by the union groups (0.28–0.31), with the multiple agreement groups demonstrating the worst performance (0.02–0.04. Increasing the number of tools from three to five did not improve performance, regardless of whether the multiple agreement or union strategies were used.

To determine the overall performance rank of each single caller and combination strategy in detecting DELs and INSs. We calculated the combined precision score (cPr), combined recall score (cRc), and combined F1 score (cF1). The values for all four datasets were integrated and calculated using a micro average (Table 4). Fig 6 also presents the performance across all callers based on the integrated values from all sample sets.

For DELs, the union strategy outperformed most single algorithms (except Manta) and multiple agreement combinations in terms of F1, achieving similar F1 scores to the commercial software DRAGEN (Fig 6 and Table 4). However, while multiple agreement strategies exhibited superior precision compared to union combinations and DRAGEN, their recall was unsatisfactory.

For INSs, the overall performance of all single callers and combination strategies was poor, with F1 scores ranging from 0.01 to 0.30, compared to DRAGEN (0.53). In terms of precision, all single callers and combination strategies performed better than DRAGEN; however, their recall was inadequate. Among all non-commercial software, Manta had the highest F1 score (0.30), second only to DRAGEN. Among the combination strategies, the union strategy achieved an F1 score comparable to Manta. These callers and combinations exhibited moderate to high precision in detecting INSs, but their overall performance was poor mainly due to the low recall rate (Fig 6 and Table 4).

Overall, increasing the number of tools from three to five did not improve performance, regardless of whether the multiple agreement or union strategies were used. However, using combination strategies did enhance performance compared to using most single callers alone.

## Discussion

SVs can have a substantial phenotypic impact, disrupting gene function, regulation, or modifying gene dosage. Their significance in medicine and molecular biology has been underscored by recent studies [32]. The role of SVs in the pathogenicity of genetic diseases is increasingly recognized, and there is a possibility that screening for SVs may become routine in clinical settings in the future.

According to our study, relying on a single algorithm for SV detection may pose the risk of limited SV size and type detection, along with low F1 score. While our multiple agreement strategy, which focuses on consensus among algorithms, tends to yield higher precision at the

**Table 3. Performance of different combination strategies in detecting DELs and INSs.**

| HG002 | | | | | | | |
|---|---|---|---|---|---|---|---|
| **SV caller** | **DEL** | **TP** | **FP** | **FN** | **Recall** | **Precision** | **F1** |
| DRAGEN | 4,787 | **3,425** | 1,362 | **2,039** | **0.63** | 0.72 | **0.67** |
| III-multiple agreement | 3,087 | <u>2,554</u> | **533** | <u>2,910</u> | <u>0.47</u> | **0.83** | 0.60 |
| V-multiple agreement | 3,565 | 2,638 | 927 | 2,826 | 0.48 | 0.74 | <u>0.58</u> |
| III-union | 5,353 | 3,380 | 1,973 | 2,084 | 0.62 | 0.63 | 0.62 |
| V-union | 5,605 | 3,406 | <u>2,199</u> | 2,058 | 0.62 | <u>0.61</u> | 0.62 |
| **SV Caller** | **INS** | **TP** | **FP** | **FN** | **Recall** | **Precision** | **F1** |
| DRAGEN | 3,403 | **2,426** | <u>977</u> | **4,855** | **0.33** | <u>0.71</u> | **0.45** |
| III-multiple agreement | 162 | <u>150</u> | **12** | <u>7,131</u> | <u>0.02</u> | **0.93** | <u>0.04</u> |
| V-multiple agreement | 162 | <u>150</u> | **12** | <u>7,131</u> | <u>0.02</u> | **0.93** | <u>0.04</u> |
| III-union | 1,799 | 1,387 | 412 | 5,894 | 0.19 | 0.77 | 0.31 |
| V-union | 1,799 | 1,387 | 412 | 5,894 | 0.19 | 0.77 | 0.31 |
| HG00514 | | | | | | | |
| **SV caller** | **DEL** | **TP** | **FP** | **FN** | **Recall** | **Precision** | **F1** |
| DRAGEN | 6,403 | **4,790** | 1,613 | **3,739** | **0.56** | 0.75 | **0.64** |
| III-multiple agreement | 3,125 | <u>2,886</u> | **239** | <u>5,643</u> | <u>0.34</u> | **0.92** | <u>0.50</u> |
| V-multiple agreement | 3,988 | 3,130 | 858 | 5,399 | 0.37 | 0.78 | <u>0.50</u> |
| III-union | 5,842 | 4,269 | 1,573 | 4,260 | 0.50 | 0.73 | 0.59 |
| V-union | 6,234 | 4,340 | <u>1,894</u> | 4,189 | 0.51 | <u>0.70</u> | 0.59 |
| **SV Caller** | **INS** | **TP** | **FP** | **FN** | **Recall** | **Precision** | **F1** |
| DRAGEN | 5,100 | **4,654** | <u>446</u> | **7,679** | **0.38** | <u>0.91</u> | **0.53** |
| III-multiple agreement | 132 | <u>129</u> | **3** | <u>12,204</u> | <u>0.01</u> | **0.98** | 0.02 |
| V-multiple agreement | 132 | <u>129</u> | **3** | <u>12,204</u> | <u>0.01</u> | **0.98** | 0.02 |
| III-union | 2,277 | 2190 | 87 | 10,143 | 0.18 | 0.96 | 0.30 |
| V-union | 2,277 | 2190 | 87 | 10,143 | 0.18 | 0.96 | 0.30 |
| HG00733 | | | | | | | |
| **SV caller** | **DEL** | **TP** | **FP** | **FN** | **Recall** | **Precision** | **F1** |
| DRAGEN | 5,895 | **4,896** | 999 | **3,798** | **0.56** | 0.83 | **0.67** |
| III-multiple agreement | 3,154 | <u>2,894</u> | **260** | <u>5,800</u> | <u>0.33</u> | **0.92** | <u>0.49</u> |
| V-multiple agreement | 4,043 | 3,158 | 885 | 5,536 | 0.36 | 0.78 | 0.50 |
| III-union | 6,013 | 4,336 | 1,677 | 4,358 | 0.50 | 0.72 | 0.59 |
| V-union | 6,426 | 4,413 | <u>2,013</u> | 4,281 | 0.51 | <u>0.69</u> | 0.58 |
| **SV Caller** | **INS** | **TP** | **FP** | **FN** | **Recall** | **Precision** | **F1** |
| DRAGEN | 5,313 | **4,868** | <u>445</u> | **7,627** | **0.39** | <u>0.92</u> | **0.55** |
| III-multiple agreement | 136 | <u>135</u> | **1** | <u>12,360</u> | <u>0.01</u> | **0.99** | 0.02 |
| V-multiple agreement | 136 | <u>135</u> | **1** | <u>12,360</u> | <u>0.01</u> | **0.99** | 0.02 |
| III-union | 2,293 | 2,218 | 75 | 10,277 | 0.18 | 0.97 | 0.30 |
| V-union | 2,293 | 2,218 | 75 | 10,277 | 0.18 | 0.97 | 0.30 |
| NA19240 | | | | | | | |
| **SV caller** | **DEL** | **TP** | **FP** | **FN** | **Recall** | **Precision** | **F1** |
| DRAGEN | 6,897 | **5,775** | <u>1,122</u> | **4,390** | **0.57** | 0.84 | **0.68** |
| III-multiple agreement | 3,694 | <u>3,418</u> | **276** | <u>6,747</u> | <u>0.34</u> | **0.93** | <u>0.49</u> |
| V-multiple agreement | 4,594 | 3,723 | 871 | 6,442 | 0.37 | 0.81 | 0.50 |
| III-union | 6,751 | 5,102 | 1,649 | 5,063 | 0.50 | 0.76 | 0.60 |
| V-union | 7,100 | 5,176 | 1,924 | 4,989 | 0.51 | <u>0.73</u> | 0.60 |
| **SV Caller** | **INS** | **TP** | **FP** | **FN** | **Recall** | **Precision** | **F1** |

*(Continued)*

**Table 3.** (Continued)

| | | | | | | | |
|---|---|---|---|---|---|---|---|
| DRAGEN | 6,144 | **5,574** | <u>570</u> | **8,579** | **0.39** | <u>0.91</u> | **0.55** |
| III-multiple agreement | 135 | <u>131</u> | **4** | <u>14,022</u> | <u>0.01</u> | **0.97** | <u>0.02</u> |
| V-multiple agreement | 135 | <u>131</u> | **4** | <u>14,022</u> | <u>0.01</u> | **0.97** | <u>0.02</u> |
| III-union | 2,481 | 2,366 | 115 | 11,787 | 0.17 | 0.95 | 0.28 |
| V-union | 2,481 | 2,366 | 115 | 11,787 | 0.17 | 0.95 | 0.28 |

FN: false negative; FP: false positive; TP: true positive; DEL: deletion. INS: insertion. The best results are highlighted in bold and the worst results are highlighted with underlines.

cost of lower recall, employing a union combination enhanced recall and F1 scores for DELs compared to both single algorithms and the multiple agreement approach. However, most callers show poor F1 scores in INS detection due to low recall rates. Specifically, DELLY (0.02) and GRIDSS (0.01) exhibited very poor F1 scores, while Manta (0.30) were the top performers among all non-commercial tools. The union strategy, which includes Manta, consequently achieved a comparable F1 score for INS detection, as illustrated in (Fig 6). This underscores the importance of choosing well-performing callers tailored to specific types and sizes of SVs for different clinical scenarios.

As for the concern of FP variants that may be falsely related to a phenotype. We were particularly interested in FPs that are consistently detected by multiple callers. We analyzed the FPs called by at least two algorithms in the HG002 dataset and compared them with DRAGEN. We differentiated FPs identified only by the multi-agreement strategy (M), only by DRAGEN

**Table 4. Combined performance of single caller and different combination strategies in detecting DELs and INSs.**

| SV caller | DEL | TP | FP | FN | Recall | Precision | F1 |
|---|---|---|---|---|---|---|---|
| DRAGEN | 23,982 | **18,886** | 5,096 | <u>13,966</u> | **0.57** | 0.79 | **0.66** |
| Manta | 17,305 | 14,878 | 2,427 | 17,974 | 0.45 | 0.86 | 0.59 |
| DELLY | 18,136 | 12,340 | 5,796 | 20,512 | 0.38 | 0.68 | 0.48 |
| GRIDSS | 11,152 | 10,350 | **802** | 22,502 | 0.32 | **0.93** | 0.47 |
| LUMPY | 14,359 | 9,663 | 4,696 | 23,189 | 0.29 | <u>0.67</u> | 0.41 |
| SvABA | 8,118 | <u>6,829</u> | 1,289 | **26,023** | <u>0.21</u> | 0.84 | <u>0.33</u> |
| III-multiple agreement | 13,060 | 11,752 | 1,308 | 21,100 | 0.36 | 0.90 | 0.51 |
| V-multiple agreement | 16,190 | 12,649 | 3,541 | 20,203 | 0.39 | 0.78 | 0.52 |
| III-union | 23,959 | 17,087 | 6,872 | 15,765 | 0.52 | 0.71 | 0.60 |
| V-union | 25,365 | 17,335 | <u>8,030</u> | 15,517 | 0.53 | 0.68 | 0.60 |
| **SV caller** | **INS** | **TP** | **FP** | **FN** | **Recall** | **Precision** | **F1** |
| DRAGEN | 19,960 | **17,522** | <u>2,438</u> | **28,740** | **0.38** | <u>0.88</u> | **0.53** |
| Manta | 8,891 | 8,202 | 689 | 38,060 | 0.18 | 0.92 | 0.30 |
| DELLY | 547 | 526 | 21 | 45,736 | <u>0.01</u> | 0.96 | 0.02 |
| GRIDSS | 333 | <u>323</u> | **10** | <u>45,939</u> | <u>0.01</u> | **0.97** | <u>0.01</u> |
| III-multiple agreement | 565 | 545 | 20 | 45,717 | <u>0.01</u> | 0.96 | 0.02 |
| V-multiple agreement | 565 | 545 | 20 | 45,717 | <u>0.01</u> | 0.96 | 0.02 |
| III-union | 8,850 | 8,161 | 689 | 38,101 | 0.18 | 0.92 | 0.30 |
| V-union | 8,850 | 8,161 | 689 | 38,101 | 0.18 | 0.92 | 0.30 |

FN: false negative; FP: false positive; TP: true positive; DEL: deletion. INS: insertion. The best results are highlighted in bold and the worst results are highlighted with underlines.

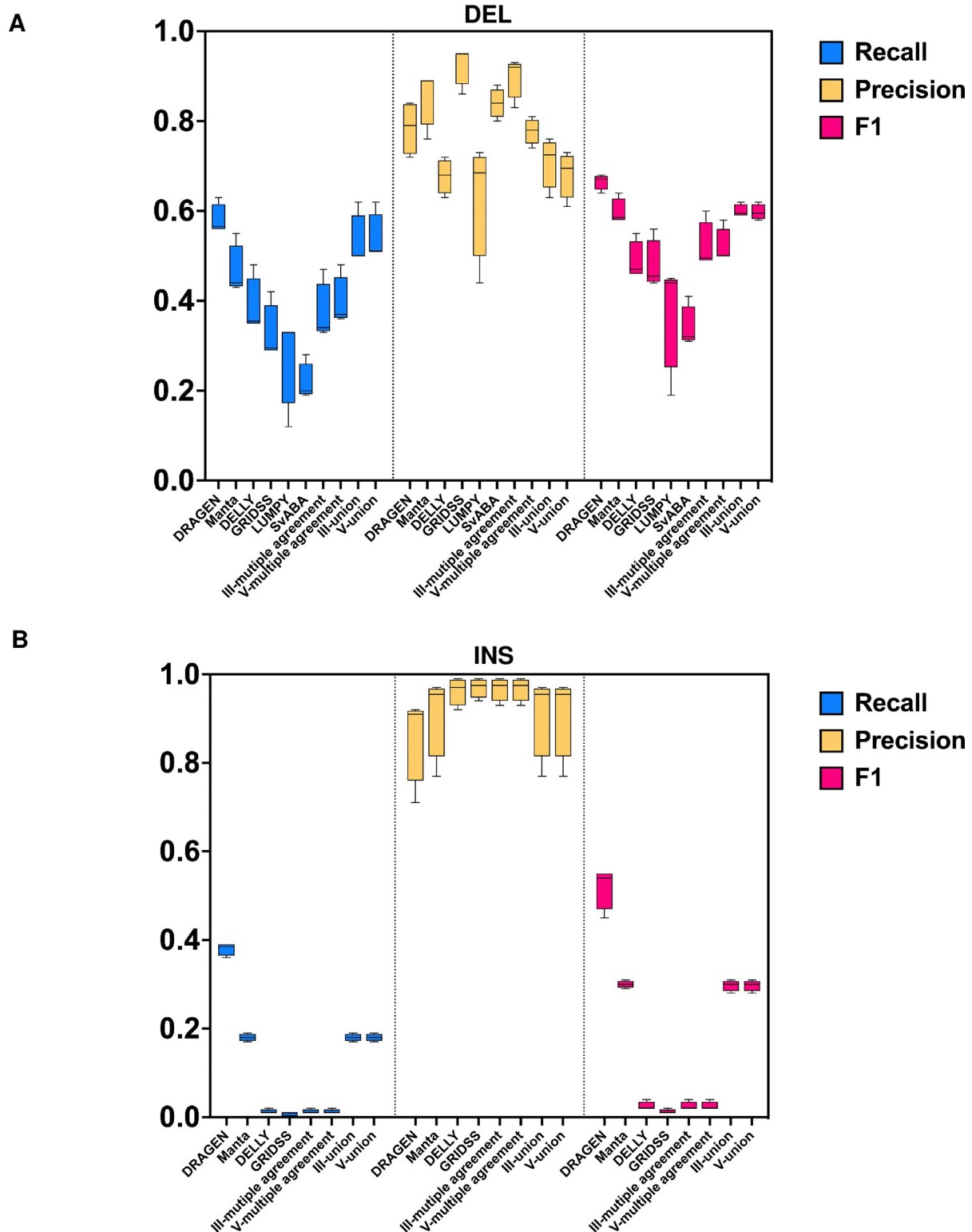

**Fig 6. Combination performance of each single caller and combination strategy in detecting DELs and INSs.** The maximum, minimum, and macro average of recall, precision, and F1 score are based on the integrated values from all sample sets.

(D), and those identified by both (Both), the FP SV size displaying the median and interquartile range were shown as S4 Fig. We found that the majority of FPs are smaller than $10^3$ base pairs. For DEL FPs, the M group had a median SV length of 326 bp, while the D group had a median of 120 bp. As for INSs, due to the poor recall of most tools, the number of FPs was quite low, with D group INS FPs having a median SV length of 81 bp. Considering that most short reads range from 300 to 500 bp in length, we believe they should be capable of accurately sequencing variants of this small size. Each tool has its own underlying principles and parameters for detecting SVs, as outlined in their respective publications, many employ hard filters—such as thresholds for variant quality, and read depth—to reduce FPs. However, care is needed to balance filtering to avoid discarding true positives. In clinical practice, post-processing tasks like filtering target genes using a virtual panel, along with additional validation through the Integrative Genomics Viewer (IGV) or methods such as PCR, are commonly used. Therefore, weighing these considerations, we believe that achieving a high recall rate to minimize FN enhances the utility of these findings in clinical settings.

Several studies have also shown that the occurrence of FPs in short-read sequencing for SV identification can be influenced by various factors, affecting overall precision. For example, Daniel L. Cameron et al. assessed the impact of sequence context and event size on the precision of 10 SV callers, evaluating these tools using simulation data and well-referenced cell lines. Their study included several tools that we also selected, such as Manta, DELLY, GRIDSS, and LUMPY. They identified several major factors that impact the precision: 1) The proximity of SNVs or indels near the SV breakpoints. When multiple flanking variants are present, false positive rates increase. 2) SVs located in low-complexity and simple tandem repeat regions are prone to higher false positive rates. SVs within LTR, LINEs, and SINEs regions also show reduced precision, leading to more false positives. 3) The size of the event also plays a role; they indicated SVs smaller than 100 bp are particularly challenging, whereas deletions between 300 to 500 bp tend to achieve higher precision.

SV detection tools based on second-generation sequencing short read data have continued to advance in recent years. Recently, Gaoyang Li et al. published PanSVR [33], which enhances SV calling by incorporating a pan-genome SV reference and read re-alignment. By leveraging SNPs and INDELs in variable number tandem repeat (VNTR) and short tandem repeat (STR) regions, PanSVR improves alignment accuracy in these regions. Furthermore, with the integration of pan-genome reference information, PanSVR enhances recall for long insertions, showcasing the advantages of a pan-genome-based approach. In their benchmark study, PanSVR achieved higher F1 scores than Manta and Delly for detecting both DELs and INSs. Moreover, Ramesh Rajaby and colleagues developed an INS caller, INSurVeyor [34], which improves the sensitivity of INS calling from short-read sequencing data, performing even better than Manta. This highlights opportunities to integrate better-performing algorithms tailored to low-complexity regions and specific SV types, like using PanSVR or INSurVeyor in combination with union strategies for enhanced performance.

Additionally, research on applying supervised learning techniques for SV detection has been growing, with notable classification-based tools such as SVM2 [35], ForestSV [36], Wham [37], and Svclassify [38]. These tools employ distinct techniques and primarily utilize information extracted from RP signals to predict candidate SV regions. Recently, several machine learning (ML)-based approaches have demonstrated impressive performance. Eman A. Alzaid and colleagues developed MPRClassify [39], which uses a multi-part read alignment strategy with three independent random forest classifiers optimized to detect specific SV types in candidate SV regions. This approach relies solely on single reads, eliminating the need for paired-end read data. Experimental results indicated that single reads can effectively identify SVs without paired-end signatures, achieving performance comparable to existing methods,

such as DELLY, Manta, and Softsv [40]. Victoria Popic et al. recently developed Cue [41], which transforms sequence alignments into images that highlight SV-related signals and utilizes a stacked hourglass convolutional neural network to predict the type, genotype, and genomic location of each SV. Evaluations on both synthetic and real short-read datasets demonstrated Cue's superior performance over established methods like Manta, DELLY, LUMPY, and SvABA, particularly in detecting a variety of SV classes, including DELs, DUPs, and INVs. In another recent study, Guiwu Zhuang et al. [30] established an expert-reviewed tumor-specific clinically relevant SV call set, evaluating several algorithms for SV detection, and developed a random-forest-based decision model to improve the precision. Building on these research foundations, it is worth exploring the potential of integrating the union combination strategy with an ML-based decision model in the future. For practical clinical applications, further investigation into the performance using real clinical sample sets is necessary.

Finally, when addressing SV calling, the role of long-read sequencing cannot be overlooked. Long-read sequencing (LRS) offers remarkable advantages as it can span repetitive or other problematic regions. For applications that require identifying very long or complex SVs, performing a localized long-read de novo assembly is more effective than conventional short-read mapping approaches. Multiple studies have demonstrated that long-read sequencing technologies offer superior accuracy in SV detection compared to short-read approaches. Benchmarking results, as shown in recent studies [42–44], consistently highlight the advantages of long reads in terms of sensitivity and specificity, regardless of the specific tools used. Moreover, the recent advancements in long-read technologies and the ongoing reductions in sequencing costs—led by both Oxford Nanopore and PacBio—have made long-read sequencing increasingly accessible and widespread. SV detection methods and post- processing approaches based on LRS have been developed to date, primarily relying on alignment-based methods that use manually designed heuristic rules tailored to sequencing platform characteristics [5, 44–47]. For instance, Svvalidator [46] and Kled [47] both collect signatures from CIGAR strings and split reads to validate SVs, with Kled further employing an Omni Merging Algorithm (OMA) to refine and genotype clusters efficiently through multi-threading and optimized memory usage in C++. However, manually designed heuristics can introduce noise and false positives, a limitation that ML-based approaches like SVDF [48] addressed by employing adaptive clustering and noise-filtering strategies. Integrating short-read and long-read sequencing provides opportunities for developing strategies tailored to diverse research and clinical needs.

Although our study only focused on the DELs and INSs—which are the majority but relatively simple SV types in the human genome—methods failing to detect DELs and INSs might also be less effective at identifying more complex SV types. Currently, our study evaluated the ability of various methods and combination strategies to detect DELs and INSs, providing a valuable and optimistic perspective on overall SV detection performance.

## Conclusions

In this study, we evaluated the performance of five advanced SV detection algorithms and commercial software, DRAGEN, using short-read whole-genome sequence data from the GIAB v0.6 Tier 1 benchmark set and well-referenced benchmark set from HGSVC2. We tested each algorithm independently and in various combinations, employing both multiple agreement and union strategies. The union approach achieved a higher recall rate, whereas the multiple strategy exhibited superior precision. Our analysis revealed that each algorithm had its intrinsic strengths and weaknesses, leading to variations in the types and sizes of SVs detected. The union combination strategies effectively coordinated these differences, enhancing overall performance, and achieving similar F1 scores as the commercial software DRAGEN.

## Supporting information

**S1 Fig. Distance of neighboring SVs in combination strategies.** The distribution of distances between start positions in neighboring SVs across different combination strategies. IQR: Interquartile Range.
(TIF)

**S2 Fig. Runtime performance of SV callers.** Runtime performance of each tool was tested on an isolated virtual machine, Taiwania 3, a high-performance computing platform featuring 900 compute nodes, each equipped with dual Intel Xeon Platinum 8280 processors (2.4 GHz, 28 cores per processor) and 192 GB of main memory. The system runs on CentOS 7.8 and utilizes the Slurm workload manager for resource scheduling. Taiwania 3 employs 100 Gbps InfiniBand HDR100 high-speed network connectivity, delivering a total computing power of up to 2.7 PFLOPS. We use node ngs92G, with total memory: 92 GB and total CPU cores: 14. Total CPU time indicates the overall CPU utilization, while wall time represents the real total elapsed time. Wall time can be less than the total CPU time if the process is executed efficiently, utilizing parallelization.
(TIF)

**S3 Fig. Efficacy of GRIDSS in detecting small variants compared to other SV detection tools.** The comparative effectiveness of GRIDSS in detecting small variants compared to other SV detection tools. *p<0.0001.
(TIF)

**S4 Fig. Length distribution of FP SVs.** FPs identified only by the multi-agreement strategy (M), only by DRAGEN (D), and those identified by both (Both), in the HG002 dataset. The FP SV size displaying the median and interquartile range were shown.
(TIF)

**S5 Fig. Combination performance of each single caller and combination strategy in detecting DELs and INSs using the 'neighboring SV method' and 'directly merge method'.** The maximum, minimum, and macro averages for recall, precision, and F1 score are calculated across all sample sets. Darker colors represent results from the 'directly merge method'. The 'neighboring SV method' is explained in the 'Combination strategies of multiple algorithms' section of the Methods, while the 'directly merge method' involves merging all caller results into one VCF file before calculating the F1 score.
(TIF)

**S1 Table. Types and sizes of all SVs detected by each individual algorithm in HG002.**
(DOCX)

**S2 Table. Types and sizes of all SVs detected by each individual algorithm in HG00514.**
(DOCX)

**S3 Table. Types and sizes of all SVs detected by each individual algorithm in HG00733.**
(DOCX)

**S4 Table. Types and sizes of all SVs detected by each individual algorithm in NA19240.**
(DOCX)

**S5 Table. Distribution of SVs in truth sets.**
(DOCX)

**S6 Table. Counts of neighboring SVs.**
(DOCX)

**S7 Table. Distribution of SVs detected by different combination strategies.**
(DOCX)

**S1 File. Example of the refinement process of "duplicated records".**
(DOCX)

**S2 File. Combination of neighboring SV from multiple callers.**
(DOCX)

## Acknowledgments

We express our gratitude for the technical support provided by the Taiwan Computing Cloud (TWCC) at the Taiwan National Center for High-Performance Computing (NCHC). The GPT-3.5 architecture, developed by OpenAI, polished the grammar, punctuation, sentence structure, and overall coherence.

## Author Contributions

**Conceptualization:** De-Min Duan, Jacob Shujui Hsu, Pei-Lung Chen.

**Data curation:** De-Min Duan.

**Formal analysis:** De-Min Duan.

**Methodology:** De-Min Duan, Chinyi Cheng, Yu-Shu Huang, An-ko Chung, Pin-Xuan Chen, Jacob Shujui Hsu, Pei-Lung Chen.

**Software:** De-Min Duan, Chinyi Cheng.

**Supervision:** Pei-Lung Chen.

**Validation:** De-Min Duan.

**Visualization:** De-Min Duan.

**Writing – original draft:** De-Min Duan, Yu-An Chen.

**Writing – review & editing:** De-Min Duan, Yu-An Chen.

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
