## [Decision Letter · Decision Letter 0]

28 May 2024

PONE-D-24-11846Comparisons of performances of Structural Variants Detection Algorithms in Solitary or Combination StrategyPLOS ONE

Dear Dr. Duan,

Thank you for submitting your manuscript to PLOS ONE. After careful consideration, we feel that it has merit but does not fully meet PLOS ONE’s publication criteria as it currently stands. Therefore, we invite you to submit a revised version of the manuscript that addresses the points raised during the review process.

Please address the reviewers' comments carefully as they have reaised some serious questions. Please explain why the need for another SV calling benchmark using existing datasets. I also noticed that you have not included any machine learning based algorithms. 

We look forward to receiving your revised manuscript.

Kind regards,

Achraf El Allali, PhD

Academic Editor

PLOS ONE

Journal Requirements:

3. In the online submission form, you indicated that "For any further requests and questions, the data used and/or analyzed during the current study are available from the corresponding author upon reasonable request."

6. We notice that your supplementary tables are included in the manuscript file. Please remove them and upload them with the file type 'Supporting Information'. Please ensure that each Supporting Information file has a legend listed in the manuscript after the references list.

**Additional Editor Comments:**

Please address the reviewer's comments carefully. Please explain why the need for another benchmarking tests. I also noticed that you have not included any machine learning based algorithms.

Reviewers' comments:

Reviewer's Responses to Questions

**Comments to the Author**

1. Is the manuscript technically sound, and do the data support the conclusions?

Reviewer #1: No

Reviewer #2: Partly

2. Has the statistical analysis been performed appropriately and rigorously? 

Reviewer #1: No

Reviewer #2: Yes

3. Have the authors made all data underlying the findings in their manuscript fully available?

Reviewer #1: Yes

Reviewer #2: Yes

4. Is the manuscript presented in an intelligible fashion and written in standard English?

Reviewer #1: Yes

Reviewer #2: Yes

5. Review Comments to the Author

Reviewer #1: The manuscript describes a benchmark experiment to detect structural variants from short reads. It uses the HG002 benchmark dataset of the Genome in a bottle project as gold standard. Due to lack of information in the gold standard or lack of events in some of the evaluated tools, this benchmark only takes into account deletions between 50 and 10kbp.

Although I understand that reviews for PloS One should not be emphasized in novelty, I think benchmarking of SVs with GIAB data is one extreme case. I do not see a point on publishing over and over the same benchmark experiment with the exact same gold standard. Just by a simple google search with the words “benchmark structural variants detection”, I obtained the following papers:

- Zook et al., 2020. https://doi.org/10.1038/s41587-020-0538-8

- Sarwal et al., 2022. https://doi.org/10.1093/bib/bbac492

- Fang et al., 2024. https://doi.org/10.1186/s12967-024-04865-w

- Cameron et al., 2019. https://doi.org/10.1038/s41467-019-11146-4

The two more recent papers are not even cited in the literature. The main issue of making all benchmarking experiments on only one dataset as gold standard is that the development of the different tools is likely to be biased towards that particular dataset, which is likely to produce overfitting. Moreover, it is at least strange that even having an almost perfect diploid assembly for this dataset, there are still no gold-standard calls for inversions and duplications. I think that the authors should come up with at least one benchmark experiment different than GIAB, to really assess the accuracy of the different tools in cases different than those probably used to tune these tools. One source to at least improve the genetic diversity of the samples is to perform benchmarking using the datasets generated by the HGSVC2 consortium (see https://doi.org/10.1038/s41467-018-08148-z and https://doi.org/10.1126/science.abf7117)

I have some additional comments, as follows.

Lines 43 – 45, and 466 - 469. It is not clear which is the evidence to support the statements made in these lines of the abstract and the conclusions. They seem to be based solely on the number of variants produced by each tool, before the assessment using the gold-standard. In particular it is impossible to draw conclusions about translocations without a gold standard to assess the accuracy of these events.

Lines 52-56: Taking into account that the decisions regarding variant detection and interpretation have been rather conservative, I really doubt that a method in which the false positives of all tools are retained would be the recommended method for diagnosis in clinical settings.

Lines 94 – 96. This statement is less true each day. Both PacBio and Nanopore have made important cost reductions and improvements in quality, reducing the cost difference with Illumina. Taking into account that SV calling with long reads has a clearly superior accuracy, compared to SV calling with short reads, I would at least discuss if it is really worth to keep considering Illumina for SV detection in clinical settings.

Lines 99 – 104. Although the issue with annotation of SVs is true, the paper does not address this issue. I recommend removing these lines, and instead writing a reduced version of what is currently on the discussion, providing a brief description of the algorithms implemented in the SV calling tools.

Lines 157 – 168: The procedures to merge results of the different tools need more clarity. What happen if for two events separated 1Kbp there is one intermediate event exactly in the middle?. It is not clear which percentage of (reciprocal) intersection is required to call two events the same. It is also not clear how the 500bp threshold was chosen. The merging called intersection is not really an intersection of the methods. It is more like a two out of n kind of union. It should not be called intersection in part because it does not make sense that the intersection of five methods resulted in more events than the intersection between three methods.

Lines 180 – 184. The procedure to call a variant true positive has the same issues as the procedure to merge calls. The 500bp limit looks arbitrary, especially if no reciprocal overlap is required. Moreover it looks too relaxed for relatively small events (lets say less than 150 bp), which are a large percentage of the total events.

Lines 271 – 274. It does not make sense to restrict the benchmark only to deletions just because two methods do not call insertions. Benchmarking of insertions should be included for the methods that predict insertions. Because comparing true and predicted insertions is more challenging, compared to deletions, it is important to define and present clearly specific rules to perform both merging and intersection of insertions.

Lines 313- 316. I would not say that the unsatisfactory results are compared to Dragen. I would say that the results are unsatisfactory overall.

Line 337. It is not surprising that the maximum is 500 if that correspond to the threshold set by the method. Again, it looks too relaxed for relatively small variants.

Lines 461 – 463. I do not see evidence of anything different or special in this work to say that it was focused on clinical applicability. There is no information on a disease or condition that could be related to the SVs detected on HG002. The authors should clarify what is their meaning of “clinical applicability”.

Finally, most of the current discussion could be written without the results of this study. Most of the text could be moved and summarized in the introduction. The text includes many sentences with qualitative assessments of each tool that are not related to the results, and in some cases are even contradictory. The description of the tools could be summarized in a couple of paragraphs of the introduction, avoiding a-priori qualitative assessments about the performance of the algorithms. Once the other comments are solved, especially finding additional gold standards and evaluating more types of variants, an actual discussion could be written based on the results of the new experiments.

Minor comments

- Figures and tables: The distribution between main and supplementary materials looks arbitrary. If supplementary tables are going to be embedded in the text, then they should not be called supplementary.

- Lines 150 -151. The sentence itself does not say much. I do not understand what is stringent about the filtering procedure. The filters mentioned in the text look fairly standard.

- Figures 1 and 2 should be composed to understand why the subfigures should go together. The figures look disorganized in the submitted version.

- Font types and resolution should be improved for Figures 3A and 3B

- The supplementary figure 3 has essentially the same information as Figure 2A. Moreover, it does not make sense to include bars for inversions and duplications if the dataset does not include these events.

Reviewer #2: Duan et al., assessed six SV calling algorithms, including DRAGEN, DELLY, LUMPY, DRIDSS, SvABA, and Manta. The assessment was done on HG002/NA24385 sample from Genome in a Bottle (GIAB).

Major Comments:

1. Reference Genome: The assessment was done on alignments against GRCh37, which is less complete than GRCh38. Please explain why GRCh38 was not used.

2. SV Truth Set: GIAB should have an officially released SV list. Why was this not used as the Truth set instead of defining your own?

3. Mendelian Violation Experiment: Since the sample used for the benchmark is a son from a trio, performing a Mendelian violation experiment by comparing with the parents would add value.

4. Conclusion Alignment: The conclusion does not fully align with the title "Comprehensive evaluation and characterisation of short read general-purpose structural variant calling software."

5. Translocation Assessment: Why were translocations not included in the assessment?

6. SvABA Reporting: Does SvABA report SVs in BND format? If so, how did you convert BND to other SV types?

7. Duplicate Records: Lines 147-148 mention that duplicated records will be removed. Under what conditions would a calling algorithm report duplicated records?

8. Duplication and Insertion Reporting: Calling algorithms may report duplications as insertions. How do you handle this case if you only combine SVs of the same type?

9. Size Range Limitation: Please explain why the comparison was limited to the range of 50 to 10k base pairs.

Minor Comments:

1. To be precise, replace “Jewish” with “Ashkenazim.”

2. The authors mention that long-read sequencing in clinical diagnostics is hindered by cost constraints, implying this study would cover this application. However, there is no coverage of this aspect in the manuscript.

3. Please spell out SNV and indel the first time they are used. Regarding SV, once it is defined, there is no need to define it every time it appears.

4. Clarify what you mean by “high-throughput analysis” in the third paragraph of the Background section (Line 78).

5. Replace “paired short reads” with “paired-end short reads” (Line 139).

6. Ensure consistent use of "nucleotide" or "base pair."

7. The term “least POS” is unclear (Line 161).

8. The statement “We systematically searched scientific articles in PubMed” is ambiguous. Consider removing it.

9. The Input and Output columns are identical for all callers, making them uninformative. Consider removing these columns.

10. Table S3 is redundant as its information is covered by Table S2.

6. PLOS authors have the option to publish the peer review history of their article (what does this mean?). If published, this will include your full peer review and any attached files.

Reviewer #1: No

Reviewer #2: **Yes: **Wan-Ping Lee

---

## [Author Response · Author response to Decision Letter 0]

9 Jul 2024

We greatly thank the editor and reviewers for the thoughtful comments of our manuscript. Following with all your kind suggestions, we have made improvement to our manuscript. The detailed point-by-point response to the comments are listed on the following pages. All the changes in revised manuscript are indicated by red color, and we also provide a clear version of our manuscript in the submission package. We really hope the revision will make the paper be more suitable for publication in PLOS ONE and look forward to hear from you in the near future. Your kind consideration will be highly appreciated!

---

## [Decision Letter · Decision Letter 1]

28 Aug 2024

PONE-D-24-11846R1Comparisons of performances of structural variants detection algorithms in solitary or combination strategyPLOS ONE

Dear Dr. Chen,

Thank you for submitting your manuscript to PLOS ONE. After careful consideration, we feel that it has merit but does not fully meet PLOS ONE’s publication criteria as it currently stands. Therefore, we invite you to submit a revised version of the manuscript that addresses the points raised during the review process.

We look forward to receiving your revised manuscript.

Kind regards,

Achraf El Allali, PhD

Academic Editor

PLOS ONE

Additional Editor Comments:

Please address the reviewers comments.

Reviewers' comments:

Reviewer's Responses to Questions

**Comments to the Author**

1. If the authors have adequately addressed your comments raised in a previous round of review and you feel that this manuscript is now acceptable for publication, you may indicate that here to bypass the “Comments to the Author” section, enter your conflict of interest statement in the “Confidential to Editor” section, and submit your "Accept" recommendation.

Reviewer #1: (No Response)

Reviewer #3: (No Response)

2. Is the manuscript technically sound, and do the data support the conclusions?

Reviewer #1: Partly

Reviewer #3: Yes

3. Has the statistical analysis been performed appropriately and rigorously? 

Reviewer #1: Yes

Reviewer #3: Yes

4. Have the authors made all data underlying the findings in their manuscript fully available?

Reviewer #1: Yes

Reviewer #3: Yes

5. Is the manuscript presented in an intelligible fashion and written in standard English?

Reviewer #1: Yes

Reviewer #3: Yes

6. Review Comments to the Author

Reviewer #1: The manuscript describes a benchmark experiment comparing tools to perform detection of SVs from short read data. The authors performed important changes in the manuscript to address most of my previous concerns, especially those related to the use of HG002 as the only benchmark dataset. They also fixed most empty, vague and even incorrect phrases likely produced by the use of large language models for automated writing (I recommend avoiding the use of such models). Regarding novelty, I would assume that the manuscript is interesting for the journal editors, taking into account that it was send again for another round of review.

I have the following remaining comments for improvement of the study, the last two related to the support of some of the statements included now in the discussion:

1. I do not see a point on removing variants longer than 10kbp from the analysis. The number of gold standard variants is small, and hence it will not affect importantly the calculations of recall. However, figure 2 suggest that there is a relatively large number of false positive variants in that size range. These false variants are important (especially if they are deletions) because they can be interpreted as important losses of DNA (and possibly genes) that could be falsely related to a trait. It would be great if the authors show an example of at least one of these predictions to rationalize why the software tools make mistakes in this size range.

2. I do not understand why variants with lengths smaller than 50bp are filtered out for all tools except GRIDSS. This makes unnecessarily complicated the interpretation of the number of variants produced by each tool. In figure 2, Figure 2A will look much better if small variants reported by GRIDSS are filtered out. In figure 2B it is not clear why different scales are used in the Y axis for the different tools if all tools report numbers within the same orders of magnitude (once the results of GRIDSS are fixed).

3. Referring to detection of SVs from long reads, the authors mention in the discussion that “A more comprehensive assessment of these technologies is still lacking”. First of all it is not clear compared to to what refers the term “more comprehensive”. More importantly, the statement is arguably false. All papers presenting new tools for SV detection from long reads include benchmark experiments (see dois 10.1093/gigascience/giad112, 10.1186/s13059-020-02107-y, and 10.1038/s41592-018-0001-7 for example ). Comparing to the accuracies presented in this work on the same benchmark datasets, it is clear that SV detection from long reads is much more accurate than SV detection from short reads, regardless the tool used for analysis. This should be clearly mentioned in the discussion, referring to the appropriate papers.

4. The authors mention in the discussion that the analysis of long reads takes a longer runtime compared to the analysis of short reads, referring to bwa. This is likely a wrong statement. First of all, it is not clear from the text why bwa is mentioned, taking into account that bwa would be the mapping tool and hence it should be compared to tools such as minimap. More importantly, the authors do not provide any result on computational efficiency of the different methods. Since this is actually an important outcome of every benchmark experiment, the authors should include a comparison of computational efficiency among the methods considered in this study. It is understandable that the runtime of their merging strategies will be at least the sum of the runtimes of the individual variant callers. If the authors want to make a statement comparing computational efficiency between the analysis of short and long reads, then they should provide comparisons including software to detect SVs from long reads.

I have some additional minor comments (line numbers refer to the version with control change):

Pages 15, 17 and 20: I think it is not worth to spell every single value of true positives, false positives and false negatives in the text, especially if you also have tables in the main text with the values. You could use ranges and only report the most important numbers that you want to highlight.

Line 24: Why the role of SVs is “pivotal”

Line 50: Base pairs is written twice

Line 82. A noun is missing before “… become more accessible”

Line 94: It is not clear which is the “sector” mentioned in the sentence.

Line 99: A simple sentence should not start with “Although”.

Line 105. The a-priori qualitative statement about SvABA probably needs a reference, especially taking into account that later SvABA turns out to be one of the worst tool in the comparison.

Line 147 - 149: The first sentence is written in active voice and the second sentence is written in passive voice. Either one can be use but do not mix up.

Line 161 - 163: The sentence is confusing. It may be better to explain first the issue and then the solution.

Line 201. strategies is misspelled

Line 246. Please explain in the methods how “state-of-the art” is defined and evaluated as a software quality attribute to select software tools.

Line 304. Please provide an statistical test and p-value to assess why the number of short variants reported by GRIDSS is “significant”. Please explain compared to what the number is significantly higher or smaller.

Lines 340 - 342: Given that the gold standard is a fixed dataset, the trend of false negatives can be directly inferred from the trend of true positives.

Line 378. The first sentence seems to need citing a reference

Line 387. It is worth to actually cite the GIAB study in this sentence

Line 406. “Close” is misspelled

Line 432 – 434. Again , given that the gold standard is a fixed dataset, the trend of false negatives can be directly inferred from the trend of true positives.

Line 451: Worst than what?

Line 478: The paper actually compares the insertions from Manta with those reported by DRAGEN. Please rephrase.

Line 490: Why the similarities are “striking”

Line 492: Significantly more intricate than what?. Please provide an statistical test and p-value to assess significance.

Line 533. The sentence is vague. Why the union method has “comprehensive coverage”?. Moreover, the sentence is contradictory with the results and even with the previous sentence. The previous sentence says that the performance of individual callers is “unsatisfactory due to a poor recall rate”, and then this sentence says that the individual callers have “strong performance”.

Reviewer #3: The author evaluated six classic SV detection tools on second-generation sequencing data and explored the impact of merging strategies on improving SV detection performance. Below are some suggestions regarding this work:

1. The author's combination experiment results indicate that the combination strategies indeed improved performance by increasing either recall or precision. However, while this conclusion is evident, the results of the two merging strategies still did not surpass the performance of the single tool, DRAGEN. To further enhance SV detection performance, the author should explore the feasibility of combination strategies based on DRAGEN’s SV detection results.

2. According to the methods described, the author’s validation approach for the combination strategy involves retaining the merged SVs from each VCF file and then calculating the overall TP, FP, and FN values. A more direct approach would be to merge the results into a single VCF file and then calculate the F1 score. The author might consider comparing the outcomes of these two validation methods.

3. In the results section, the extensive listing of data might obscure the main conclusions. It is recommended that the author focus on summarizing the key conclusions, present comparative data in tables, and highlight the best results in bold.

4. The discussion section devotes significant space to background and motivation, which would be better emphasized in the introduction. It is advisable to reposition this content accordingly.

5. The author is advised to refine the abstract for clarity and conciseness.

6. The low recall rate of DELLY and GRIDSS tools in detecting INS variants is unconvincing. The author should verify the data or provide a reasonable explanation based on the literature.

7. All formulas in the manuscript should be numbered for reference and citation purposes.

8. The term ‘INFOa’ on page 8 of the manuscript should be corrected to ‘INFO’.

9. The representation of recall, precision, and F1 scores should be consistent. For example, if the F1 score is 0.67, then the recall should be represented as 0.63, not as a percentage like 62.57.

7. PLOS authors have the option to publish the peer review history of their article (what does this mean?). If published, this will include your full peer review and any attached files.

Reviewer #1: No

Reviewer #3: No

---

## [Author Response · Author response to Decision Letter 1]

10 Oct 2024

We greatly thank the editor and reviewers for the thoughtful comments of our manuscript. Following with all your kind suggestions, we have made improvement to our manuscript. The detailed point-by-point response to the comments are listed on the following pages. All the changes in revised manuscript are indicated by red color, and we also provide a clear version of our manuscript in the submission package. We really hope the revision will make the paper be more suitable for publication in PLOS ONE and look forward to hear from you in the near future. Your kind consideration will be highly appreciated!

---

## [Decision Letter · Decision Letter 2]

22 Oct 2024

PONE-D-24-11846R2Comparisons of performances of structural variants detection algorithms in solitary or combination strategyPLOS ONE

Dear Dr. Chen,

Thank you for submitting your manuscript to PLOS ONE. After careful consideration, we feel that it has merit but does not fully meet PLOS ONE’s publication criteria as it currently stands. Therefore, we invite you to submit a revised version of the manuscript that addresses the points raised during the review process.

We look forward to receiving your revised manuscript.

Kind regards,

Achraf El Allali, PhD

Academic Editor

PLOS ONE

Journal Requirements:

Additional Editor Comments:

The reviewers have finished their review and have minor comments that I include here.

R1

The authors address properly my previous comments. As a final note, please make sure that the final versions of the figures have good quality and try to increase font sizes as much as possible. In particular the numbers in figure 4 are very difficult to see.

R2

1. Please conduct a thorough review of the format and text throughout the document. Specifically, the citation at line 75 should be placed before the period, and the comma usage at line 274 needs to be corrected.

2. I recommend reordering the references to align as closely as possible with the order in which they appear in the text.

3. The manuscript primarily discusses tools published before 2020; however, significant advancements have been made in SV detection tools based on second-generation sequencing data in recent years. For instance, tools like "PanSVR: Pan-genome augmented short read realignment for sensitive detection of structural variations" and "Psi-Caller: a lightweight short read-based variant caller with high speed and accuracy" represent the latest research outcomes and should be discussed more thoroughly in the text. Additionally, the discussion on third-generation sequencing data is insufficient. It is advisable to incorporate discussions on the latest tools for SV detection using third-generation sequencing data, such as "SVDF: enhancing structural variation detection from long-read sequencing via automatic filtering strategies" and "Kled: an ultra-fast and sensitive structural variant detection tool for long-read sequencing data," to enrich and enhance the discussion section.

In the spirit of what reviewer 2 have mentioned, there are some other tools that use machine learning and post processing approaches such as the following doi: 10.4172/0974-276X.1000488 and https://doi.org/10.1177/117793221989295

Since Plos one does not have a proofreading step, please check all the grammar and make sure all figures have the best quality possible.

Reviewers' comments:

Reviewer's Responses to Questions

**Comments to the Author**

1. If the authors have adequately addressed your comments raised in a previous round of review and you feel that this manuscript is now acceptable for publication, you may indicate that here to bypass the “Comments to the Author” section, enter your conflict of interest statement in the “Confidential to Editor” section, and submit your "Accept" recommendation.

Reviewer #1: All comments have been addressed

Reviewer #3: All comments have been addressed

2. Is the manuscript technically sound, and do the data support the conclusions?

Reviewer #1: Yes

Reviewer #3: Yes

3. Has the statistical analysis been performed appropriately and rigorously? 

Reviewer #1: Yes

Reviewer #3: Yes

4. Have the authors made all data underlying the findings in their manuscript fully available?

Reviewer #1: Yes

Reviewer #3: Yes

5. Is the manuscript presented in an intelligible fashion and written in standard English?

Reviewer #1: Yes

Reviewer #3: Yes

6. Review Comments to the Author

Reviewer #1: The authors address properly my previous comments. As a final note, please make sure that the final versions of the figures have good quality and try to increase font sizes as much as possible. In particular the numbers in figure 4 are very difficult to see.

Reviewer #3: The author has made revisions in response to the issues I raised, however, I still I have the following remaining comments:

1. Please conduct a thorough review of the format and text throughout the document. Specifically, the citation at line 75 should be placed before the period, and the comma usage at line 274 needs to be corrected.

2. I recommend reordering the references to align as closely as possible with the order in which they appear in the text.

3. The manuscript primarily discusses tools published before 2020; however, significant advancements have been made in SV detection tools based on second-generation sequencing data in recent years. For instance, tools like "PanSVR: Pan-genome augmented short read realignment for sensitive detection of structural variations" and "Psi-Caller: a lightweight short read-based variant caller with high speed and accuracy" represent the latest research outcomes and should be discussed more thoroughly in the text. Additionally, the discussion on third-generation sequencing data is insufficient. It is advisable to incorporate discussions on the latest tools for SV detection using third-generation sequencing data, such as "SVDF: enhancing structural variation detection from long-read sequencing via automatic filtering strategies" and "Kled: an ultra-fast and sensitive structural variant detection tool for long-read sequencing data," to enrich and enhance the discussion section.

7. PLOS authors have the option to publish the peer review history of their article (what does this mean?). If published, this will include your full peer review and any attached files.

Reviewer #1: **Yes: **Jorge Duitama

Reviewer #3: No

---

## [Author Response · Author response to Decision Letter 2]

16 Nov 2024

We greatly thank the editor and reviewers for the thoughtful comments of our manuscript. Following with all your kind suggestions, we have made improvement to our manuscript. The detailed point-by-point response to the comments are listed on the following pages. All the changes in revised manuscript are indicated by red color, and we also provide a clear version of our manuscript in the submission package. We really hope the revision will make the paper be more suitable for publication in PLOS ONE and look forward to hear from you in the near future. Your kind consideration will be highly appreciated!

---

## [Editor Report · Decision Letter 3]

20 Nov 2024

Comparisons of performances of structural variants detection algorithms in solitary or combination strategy

PONE-D-24-11846R3

Dear Dr. Chen,

We’re pleased to inform you that your manuscript has been judged scientifically suitable for publication and will be formally accepted for publication once it meets all outstanding technical requirements.

Kind regards,

Achraf El Allali, PhD

Academic Editor

PLOS ONE

Additional Editor Comments (optional):

Line 472, "may could" should be replaced by one of the two words

Line 478 you should not use 10^ but rather use the superscript in word or latex.
---

## [Editor Report · Acceptance letter]

24 Jan 2025

PONE-D-24-11846R3 

PLOS ONE

Dear Dr. Chen, 

I'm pleased to inform you that your manuscript has been deemed suitable for publication in PLOS ONE. Congratulations! Your manuscript is now being handed over to our production team.

Kind regards, 

on behalf of

Dr. Achraf El Allali 

Academic Editor

PLOS ONE